# PEAKS: Selecting Key Training Examples Incrementally via Prediction Error Anchored by Kernel Similarity

**Mustafa Burak Gurbuz**[1]  **Xingyu Zheng**[2]  **Constantine Dovrolis**[1][3]

## Abstract

As deep learning continues to be driven by ever-larger datasets, understanding which examples are most important for generalization has become a critical question. While progress in data selection continues, emerging applications require studying this problem in dynamic contexts. To bridge this gap, we pose the Incremental Data Selection (IDS) problem, where examples arrive as a continuous stream, and need to be selected without access to the full data source. In this setting, the learner must incrementally build a training dataset of predefined size while simultaneously learning the underlying task. We find that in IDS, the impact of a new sample on the model state depends fundamentally on both its geometric relationship in the feature space and its prediction error. Leveraging this insight, we propose PEAKS (Prediction Error Anchored by Kernel Similarity), an efficient data selection method tailored for IDS. Our comprehensive evaluations demonstrate that PEAKS consistently outperforms existing selection strategies. Furthermore, PEAKS yields increasingly better performance returns than random selection as training data size grows on real-world datasets. The code is available at https://github.com/BurakGurbuz97/PEAKS.

## 1. Introduction

Deep neural networks (DNNs) have achieved remarkable success in recent years, fueled in large part by the availability of massive datasets (Zhai et al., 2022; Touvron et al., 2023; Liu et al., 2024). However, scaling up training data incurs substantial costs: data collection, increased computa-

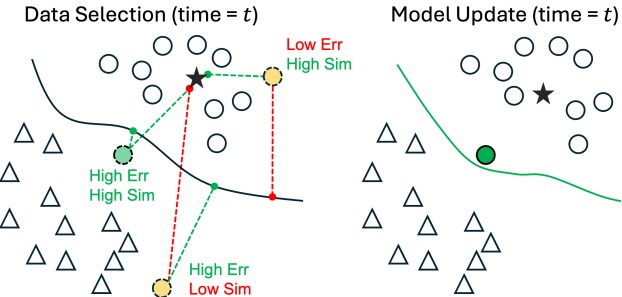

*Figure 1.* Illustration of PEAKS: The solid line denotes the decision boundary. Empty markers represent current training samples, and the star indicates the mean embedding of the circular class. During the selection step, among the three potential candidates for inclusion in the training dataset, the green circle is selected due to its high prediction error (Err) and strong similarity (Sim) to the mean embedding. The model is subsequently updated using the newly selected sample along with prior examples. This iterative process is repeated until a total of $k$ new examples are collected.

tional requirements, and environmental impact from energy consumption (Yang et al., 2023). Thus, there is a growing need to improve the data efficiency of DNNs to ensure their development remains accessible and sustainable.

This pressing need for data efficiency has sparked renewed interest in coreset selection, a well-studied problem in traditional machine learning that is now finding applications in deep learning. A common approach is static coreset selection, which identifies a representative subset of the training data in one shot using a trained model (Guo et al., 2022). However, this approach assumes an example's value remains fixed regardless of the training trajectory and model state—an assumption that holds for traditional algorithms (e.g., SVMs) but not for DNNs. Adaptive methods address this limitation by dynamically reselecting coresets throughout training (Yang et al., 2023; Killamsetty et al., 2021b). Yet both approaches require complete datasets to be available upfront and need to process the entire dataset for selection, making the selection process itself impractical as datasets grow to millions or billions of examples.

Online batch selection methods offer a more practical approach by evaluating examples in each training batch and selectively using only the most informative examples for

[1]School of Computer Science, Georgia Institute of Technology, USA [2]School of Biological Sciences, Cold Spring Harbor Laboratory, USA [3]The Cyprus Institute, Cyprus. Correspondence to: Mustafa Burak Gurbuz <mgurbuz6@gatech.edu>.

*Proceedings of the 42$^{nd}$ International Conference on Machine Learning*, Vancouver, Canada. PMLR 267, 2025. Copyright 2025 by the author(s).

updates (Nguyen et al., 2024; Wang et al., 2024). However, when data cannot be collected and shuffled beforehand, this local focus becomes problematic. Real-world data is inherently heterogeneous, with varying quality levels across sources and potential for sudden distribution shifts (Goyal et al., 2024; Delange et al., 2021). In such cases, batches may consist entirely of low-quality samples or reflect temporary distribution changes, making selections and training updates based solely on the local batch context unreliable.

As deep learning applications continue to grow, they often operate in a dynamic and restrictive context that require studying data selection in new scenarios. Consider on-device learning and personalization, where devices continuously receive user data that cannot be shuffled beforehand, is highly redundant, but needs to be filtered under resource constraints to capture high-value signals for model improvement (Moon et al., 2024). Similarly, computer vision applications relying on web-scraped data must contend with streams of images of varying quality, from high-resolution to noisy or irrelevant ones (Li et al., 2017). In such settings, while identifying promising samples is crucial, the greater challenge lies in retaining them effectively—valuable data should not be discarded after an update, as models typically require multiple exposures to learn effectively, with one-time updates potentially causing disruptive changes (e.g., large biased updates) or being overwritten in subsequent training (Toneva et al., 2019). Instead, we argue that these samples should be retained and repeatedly integrated into training, where their collective interactions over multiple passes help develop more effective representations.

To address these challenges, we propose Incremental Data Selection (IDS), a novel and pragmatic problem setting for data-efficient learning. In IDS, data arrives as a continuous stream, requiring the learner to make incremental selection decisions while training. The streaming nature of IDS ties the utility of each example to the model's evolving state, as selection choices must build upon and refine the current representation space. Unlike adaptive or online methods where selections are often temporary, IDS retains chosen examples in the training set. These samples are then mixed with newly arriving data during subsequent updates, providing stable representation learning (see **Figure** 2). Furthermore, this permanent selection mechanism incrementally constructs a coreset of previously seen examples up to a predefined size, enabling systematic study of how training data size influences overall performance.

In the next section, we formalize the IDS problem and review related work. **Section** 3 studies IDS by examining how new samples affect the network's predictions and its internal representations, providing insights into the relationship between sample utility and model state. Building on these insights, **Section** 4 presents PEAKS, an efficient algorithm

for IDS. Finally, we validate our approach through comprehensive experiments on four datasets, ranging from clean to noisy and imbalanced, to reflect real-world complexity. Our evaluation primarily focuses on fine-tuning pre-trained models with a limited data budget for complex image classification tasks, where the value of principled data selection becomes particularly evident. While our method shows its greatest impact in fine-tuning scenarios, it also provides consistent benefits when training from scratch.

## 2. Problem Formulation and Related Work

We study data selection for classification; however, the problem formulations and mathematical analysis presented in the next section are applicable to regression as well. To study incremental data selection in a principled way, this work proposes a version of IDS that captures the essential aspects of sequentially selecting data with a fixed budget. The setting considers a data source $D_{\text{source}}$ of unknown size. Access to $D_{\text{source}}$ is limited to incremental sampling, where labeled examples $(x, y) \sim D_{\text{source}}$ are observed one at a time. The goal is to select $k$ examples to train $f(x; \theta)$, such that it learns the underlying task and performs well on a clean testing set. Following standard deep learning training practices, IDS has three phases (see **Figure** 2):

**1) Initialization:** The process begins by randomly sampling $m$ examples from $D_{\text{source}}$ ($m \ll k$) to form the initial training set $T_0$. The initial model (with or without pretraining) trains on $T_0$ until convergence, resulting in $f(x; \theta_0)$. In this step, $f(x; \theta_0)$ learns basic task-relevant features before data selection begins. Next, $T_0$ grows by selecting new samples.

**2) Data Selection:** This phase alternates between example selection and model update. Consider selection step $t$:

**Example Selection:** A potential example $(x_p, y_p)$ is sampled from $D_{\text{source}} \setminus T_t$. The decision to keep this sample is made using an acquisition function $A(x_p, y_p, f_{\theta_t}; \theta_A)$, where $\theta_A$ is the parameters of this function, which is updated after each selection (e.g., a selection threshold). This process is repeated until $\delta$ new examples are selected.

**Model and Dataset Update:** After collecting $\delta$ examples, $S_{\text{new}} = \{(x_i, y_i)\}_{i=1}^{\delta}$, a training batch of size $b$ is formed by combining these new examples with $b - \delta$ older examples sampled uniformly at random[1] from $T_t$. The model is then updated using this batch, and the training set is updated as $T_{t+1} = T_t \cup S_{\text{new}}$. This process is repeated until $|T_{t+1}| = k$, forming the final set $T_{\text{end}}$ for the next phase.

**3) Final Training:** In final phase, the model is fine-tuned on the final training set $T_{\text{end}}$. This ensures the model learns all selected examples, including the most recent additions.

---

[1]Alternative sampling mechanism is discussed in **Section** D.4

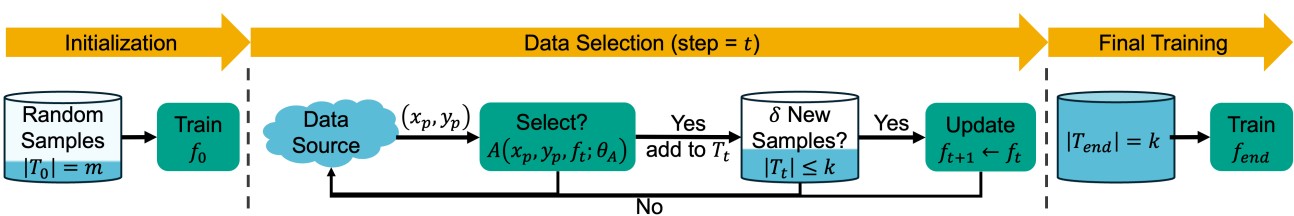

*Figure 2.* Incremental data selection consists of three phases. First, the model is trained on $m$ randomly selected examples. During the main data selection phase, new examples are iteratively selected from the data source using the acquisition function $A$. The model is updated each time $\delta$ new examples are collected. Once the data budget $k$ is reached, the model is fine-tuned on all the selected examples.

The objective of IDS is to maximize test accuracy through an acquisition function. The framework involves two hyperparameters: $k$ (data budget) and $\delta$ (selection increment). We set $\delta$ to ensure half of the total training updates ($\frac{u}{2}$) occur during data selection. For example, with $k = 1000$ and $u = 200$, we set $\delta = 10$. This approach performs consistently well and simplifies experimentation by making $k$ the only free variable. See **Section B.7** for practical considerations for efficiently scaling IDS in large-scale experiments.

### 2.1. Related Work

**Static and Adaptive Coreset Selection:** Coreset selection aims to identify a subset of the training data that allows a model to achieve performance comparable to training on the full dataset. This is typically approached in one of two ways. The static method selects a subset at once after training a model on the full dataset, then trains a new model from scratch on the reduced dataset (Paul et al., 2021; Sorscher et al., 2022; Meding et al., 2022; Coleman et al., 2020; Mindermann et al., 2022; Xia et al., 2024b; Zheng et al., 2023; Yang et al., 2024). A key limitation of static methods is their assumption that the importance of samples is independent of the model's state, as selection is performed in isolation from the training.

In contrast, adaptive methods periodically reselect subsets from the full dataset during training to remain their effectiveness (Mirzasoleiman et al., 2020; Yang et al., 2023; Killamsetty et al., 2021b;a). However, both approaches require processing the entire dataset multiple times—either during the training phase or through repeated reselection—resulting in significant computational costs. Furthermore, these approaches become inapplicable when the full dataset is not available upfront.

**Online Batch Selection:** These methods evaluate examples at the batch level, selecting only a subset of each batch for model updates (Jiang et al., 2019; Katharopoulos & Fleuret, 2018). This approach has proven particularly effective for LLM training, where it offers a practical way to reduce computational costs when batches contain a diverse mix of high- and low-value samples (Nguyen et al., 2024; Wang et al., 2024). However, when data cannot be collected and shuffled beforehand, batches may not be representative of the broader dataset distribution. In such cases, individual batches might consist primarily of low-quality or redundant samples, or reflect short-term distribution shifts, rendering local batch-level selection suboptimal.

Furthermore, these methods typically make temporary selection decisions, either discarding samples after a single update (Wang et al., 2024) in single-pass learning or allowing previously selected samples to be rejected in subsequent epochs (Jiang et al., 2019). This presents two limitations. First, in single-pass scenarios, discarding samples after one update may be suboptimal, as many models and tasks require multiple exposures to learn training examples. Second, in multi-epoch settings, despite aiming to reduce per-epoch computation, these methods typically process most of the dataset across epochs through continuous rotation and reconsideration of selected samples. In contrast, IDS constructs a training set of predefined size while enabling repeated usage of selected examples. This allows for systematic study of the relationships between dataset size, sample repetition, and model performance—a topic of growing interest given empirically observed neural scaling laws (Sorscher et al., 2022; Muennighoff et al., 2024; Zhai et al., 2022).

**Connections to Broader Learning Settings:** Our work also connects to several related research directions. The streaming nature of data selection shares similarities with streaming active learning (Settles, 2009), while the presence of noisy or imbalanced examples in $D_{\text{source}}$ relates closely to robust and long-tailed learning (Song et al., 2022; Zhang et al., 2023). Although our current formulation assumes a stationary distribution for $D_{\text{source}}$ and focuses exclusively on data efficiency, the IDS framework naturally aligns with continual learning settings where the data distribution evolves over time. While PEAKS is not designed for continual learning, its incremental update mechanism bears resemblance to replay-based strategies commonly used in that domain (Delange et al., 2021). We elaborate on these connections and important distinctions in **Appendix C**.

## 3. Analyzing Impact of New Data on Model

In IDS, after the initialization phase, the model should have improved largely from random guessing and acquired decent zero-shot performance. In line with two recent work (Dong et al., 2024; Xia et al., 2024a), we assume the network function at this point is close to the optimal parameter space, and thus the model's behavior between consecutive updates can be approximated via first-order Taylor expansion, operating in the kernel regime[2].

To effectively select examples in the IDS setting, we need to understand how incorporating a potential example $(x_p, y_p)$ into training impacts the model's predictions on validation samples, and therefore able to assess its value for future learning. Here, $y \in \mathbb{R}^C$ is the one-hot-encoded label. The model is defined as $f(x; \theta_t)$ at training time $t$, and the predicted class probabilities are given by $\sigma(f(x; \theta_t))$, where $\sigma$ represents the softmax operator, and $\sigma(f(x; \theta_t))[i]$ denotes the probability for class $i$.

Let $\Delta f_v = f(x_v, \theta_{t+1}) - f(x_v, \theta_t) \in \mathbb{R}^C$ denote the change in model output for a validation sample after one SGD update using example $x_p$. Using first-order Taylor expansion and gradient descent with learning rate $\eta$:

$$\Delta f_v = -\eta \nabla_\theta f(x_v; \theta_t)^T \nabla_\theta f(x_p, \theta_t) \cdot (\sigma(f(x_p, \theta_t)) - y_p) \tag{1}$$

The first term, $\nabla_\theta f(x_v; \theta_t)^T \nabla_\theta f(x_p, \theta_t)$, is the Neural Tangent Kernel $\mathcal{K}(x_v, x_p) \in \mathbb{R}^{C \times C}$, which characterizes how training on input $x_p$ induces changes in the model's predictions for input $x_v$ through parameter updates (Jacot et al., 2018). The second term, $(\sigma(f(x_p, \theta_t)) - y_p)$, is the gradient of the cross-entropy loss with respect to model outputs. To understand how updating on example $x_p$ affects specific logits, we examine $\Delta f_v[i]$ with the kernel entries:

$$\Delta f_v[i] = \sum_{j=1}^{C} \mathcal{K}(x_v, x_p)[i, j](y_p[j] - \sigma(f(x_p, \theta_t))[j]) \tag{2}$$

Here, $\Delta f_v[i]$ represents the change in the $i$-th logit of the validation example, which depends on both the kernel interaction $\mathcal{K}(x_v, x_p)[i, j]$ and the logit gradient for each class $j$. The latter measures how much the model's predicted probability distribution deviates from the class distribution, which we refer to as the prediction error. Putting together, the change in logit $i$ depends on logit gradients for all classes weighted by their kernel coupling $\mathcal{K}(x_v, x_p)[i, j]$, thus influenced by both the geometric relationship between examples in parameter space and the model's prediction error.

When selecting beneficial training examples, a natural objective is to maximize logit increases for correct classes while

encouraging logit decreases for incorrect classes. This motivates our scoring function for example $x_p$:

$$\Delta(x_p, x_v) = \Delta f_v[c_v] - \sum_{i \neq c_v} \Delta f_v[i] \tag{3}$$

where $c_v := y_v$. Through eq. (2), we can see that the high scoring examples should satisfy two criteria: they should have significant prediction errors (indicating potential for improvement), and gradient patterns that align with validation examples. The prediction error term identifies examples where the model needs improvement, while kernel coupling term naturally emphasizes examples that are geometrically related in the model's parameter space. Our derivation provides a principled way to analyze how training on a new example influences the model's predictions.

### 3.1. Simplification for Tractable Algorithm

Computing $\Delta f_v[i]$ requires calculating the Jacobians of all logits with respect to the network parameters. With modern DNNs containing millions of parameters and hundreds of output classes, this quickly becomes intractable. Therefore, we consider the following simplification.

**Last Layer Training**: We assume updates are limited to the classification layer $L(x) = W\phi(x)$, where $W \in \mathbb{R}^{d \times C}$ and $\phi(x)$ is the penultimate layer representation ($L(x)$ relates to the network function: $f(x; W) = L(\phi(x; \theta_{1:l-1}))$). This approximation is justified by empirical findings that later layers undergo more significant changes during training while early layers remain relatively stable (Yosinski et al., 2014), whether using a pretrained model or after sufficient initial training. We focus on how training on $(x_p, y_p)$ affects validation samples of the same class (i.e., where $c_v = y_p$). Under last-layer training, $\mathcal{K}(x_v, x_p)$ factorizes as $\phi(x_v)^T \phi(x_p)$ when $i = j$, and is zero elsewhere. The kernel simplifies to:

$$\mathcal{K}(x_v, x_p)[i, j] = \begin{cases} \phi(x_v)^T \phi(x_p), & \text{if } i = j \\ 0, & \text{if } i \neq j \end{cases} \tag{4}$$

With this, our objective $\Delta(x_p, x_v)$ becomes:

$$(\phi(x_v)^T \phi(x_p))(1 - \sigma(f(x_p, \theta_t))[y_p]) + \tag{5}$$

$$(\phi(x_v)^T \phi(x_p)) \sum_{i \neq y_p} \sigma(f(x_p, \theta_t))[i] \tag{6}$$

which approximates the net positive effect of training on $x_p$ for validating example $x_v$. This is simply:

$$\Delta(x_p, x_v) = E(x_p)(\phi(x_v)^T \phi(x_p)) \tag{7}$$

where $E(x_p)$ captures the model's prediction error:

$$E(x_p) = (1 - \sigma(f(x_p, \theta_t))[y_p]) + \sum_{i \neq y_p} \sigma(f(x_p, \theta_t))[i] \tag{8}$$

---

[2]These assumptions require a larger initial set $T_0$ when training from scratch, compared to fine-tuning a pre-trained model.

Therefore, taking the expectation over all validation examples of class $y_p$:

$$\mathbb{E}_{x_v \sim D_v^{y_p}}[\Delta(x_p, x_v)] = E(x_p) \left\langle \phi(x_p), \frac{\sum_{x_v \in D_v^{y_p}} \phi(x_v)}{|D_v^{y_p}|} \right\rangle \tag{9}$$

This shows that $x_p$ improves predictions based on its error $E(x_p)$ and similarity to the class mean embedding. Interestingly, these terms compete: maximizing $E(x_p)$ aligns with the EL2N score in (Paul et al., 2021), while the feature product term captures how close $x_p$'s features are to the class prototype (mean embedding) (Sorscher et al., 2022). Our method suggests using the product of these scores rather than considering them independently.

We argue that combining prediction error with embedding similarity enables more effective selection of informative examples. This product emphasizes examples that are both challenging (i.e., the model currently predicts them poorly) and representative (i.e., their embeddings align with the class prototype). Such examples are likely to induce beneficial updates that generalize across many other typical samples in the same class.

In contrast, relying solely on prediction error could prioritize high-error examples that may be mislabeled, atypical, or out-of-distribution—particularly common in real-world data. On the other hand, selecting purely based on embedding similarity tends to favor overly typical or already well-learned examples, filtering out noise but also discarding hard cases that could drive model improvement. By integrating both criteria, our scoring function selectively promotes hard-but-clean examples.

### 3.2. Eliminating the Need for a Validation Set

Our scoring function $\mathbb{E}_{D_v^{y_p}}[\Delta(x_p, x_v)]$ requires validation examples to compute class mean embeddings. However, the weight vector $W_{[:,y_p]}$ already serves as a learned prototype for class $y_p$, approximating the true class prototype after initial training on randomly selected data. Therefore, the logit $f(x, \theta_t)[y_p] = W_{[:,y_p]}^T \phi(x)$ measures feature alignment with this prototype.

Using this geometric interpretation for the weight vectors, we can approximate our score using $x_p$'s features and logits:

$$\mathbb{E}_{D_v^{y_p}}[\Delta(x_p, x_v)] \approx E(x_p) \left\langle \phi(x_p), W_{[:,y_p]} \right\rangle \tag{10}$$
$$= E(x_p) f(x_p, \theta_t)[y_p] \tag{11}$$

We empirically validate the quality of these approximations in Appendix D.1, showing strong agreement with the exact scoring function.

## 4. PEAKS for Incremental Data Selection

While traditional coreset selection require access to the full dataset, the scoring function from our theoretical analysis offers a readily usable way to evaluate examples incrementally. Based on this insight, we propose PEAKS (**P**rediction **E**rror **A**nchored by **K**ernel **S**imilarity), which leverages theoretical insights from **Section** 3 that establish a sample's utility depends on both geometric relationships and errors. We introduce two variants: PEAKS-V, which requires a validation set and involves periodic computation of class means (eq. (9)), and PEAKS, which eliminates this requirement (eq. (11)). Our experiments primarily focus on PEAKS, which provides an efficient solution for IDS (see **Figure** 1).

### 4.1. Data Selection with PEAKS and PEAKS-V

Consider a potential example $(x_p, y_p)$ presented at selection step $t$. As discussed in **Section** 3, $\mathbb{E}_{D_v^{y_p}}[\Delta(x_p, x_v)]$ estimates how much a sample improves predictions for class $y_p$. However, this score is inherently class-dependent, and comparing it directly across classes can be misleading. For instance, a change of $\Delta$ in the activation of an output neuron might significantly improve predictions for one class while having minimal impact for another class. Since the IDS scenario does not allow comparison across a pool of examples with different classes, we propose a simple solution: weighting $\mathbb{E}_{D_v^{y_p}}[\Delta(x_p, x_v)]$ by $\frac{1}{c_{y_p}(t)}$, where $c_{y_p}(t)$ represents the number of samples already selected from class $y_p$ at time $t$. Thus, for a sample, we define the score $s(x_p, y_p)$ as:

$$s(x_p, y_p) = \frac{\mathbb{E}_{D_v^{y_p}}[\Delta(x_p, x_v)]}{c_{y_p}(t)} \tag{12}$$

$\mathbb{E}_{D_v^{y_p}}[\Delta(x_p, x_v)]$ is calculated via eq. (11) or (9) for PEAKS and PEAKS-V, respectively. This adjustment avoids extremely imbalanced selection and provides consistent benefits on balanced and imbalanced datasets (see **Section** D.5).

### 4.2. Score-based Dynamic Selection Criterion

Selection decisions require a dynamic threshold that adapts to evolving score distributions during model training[3]. To achieve this, we maintain a cache $\mathcal{C}$ of scores from recently seen examples and make decisions based on percentile rankings. For a sample $(x_p, y_p)$ with score $s(x_p, y_p)$, we compute its percentile rank within $\mathcal{C}$. Given a selection ratio $p\%$, the acquisition function $A(x_p, y_p, f_{\theta_i}; \mathcal{C})$ is:

$$\text{Percentile}(s(x_p, y_p), \mathcal{C}) \geq 100 - p \tag{13}$$

Otherwise, the sample is discarded. Every $\tau$ updates, the recent score cache $\mathcal{C}$ is cleared. For PEAKS-V, we also recompute class mean embeddings. This dynamic selection

---

[3]The evolution of selection scores is analyzed in **Section** D.2.

criterion is adapted for all score-based methods in our experiments, where each method defines its own acquisition function $A(x_p, y_p, f_\theta; \mathcal{C})$ with two hyperparameters: the selection rate $p\%$ and the refresh period $\tau$. The complete IDS procedure is formalized in **Algorithm** 1.

## 5. Experimental Results

Our experiments spans four datasets: CIFAR100, Food101 (Bossard et al., 2014), Food101-N (Lee et al., 2018), and WebVision (Li et al., 2017), chosen to assess IDS across varying data quality. CIFAR100 and Food101 serve as balanced, relatively clean datasets, while Food101-N (a distinct dataset from Food-101 collected from different sources) introduces class imbalance and label noise. WebVision represents a large-scale real-world scenario. Dataset details and selection criteria are discussed **Section** B.5.

We compare PEAKS against eight baselines. Three variants utilize penultimate layer embeddings—Easy (Welling, 2009), Hard (Sorscher et al., 2022), and Moderate (Xia et al., 2023)—selecting examples based on their cosine similarity (highest, lowest, and intermediate, respectively) to class mean embeddings. These are computed either via validation set or approximated using output layer weights as described in **Section** 3.2. The remaining baselines include EL2N (error magnitude), GraNd (gradient norm) (Paul et al., 2021), uncertainty-based selection (prediction confidence) (Guo et al., 2022), wrong and low confidence and uniform random selection. The rationale behind baseline selection and details are presented in B.6.

We set the selection rate $p\%$ to 20 across all experiments, while $\tau$ is configured to ensure 10 cache refresh operations occur during the data selection phase, unless otherwise specified. While PEAKS algorithm's derivation assumes training only the classification layer to derive an efficient selection method, our experiments train the entire architecture. Additional experimental details are provided in their respective subsections and comprehensively documented in **Section** B.

### 5.1. Training from Scratch Results

We first perform IDS by training a ResNet-18 from scratch. In this setting, a larger initial training set size $m$ is necessary to ensure the model achieves sufficient performance for meaningful selection decisions, as both PEAKS and other baseline methods rely on model outputs being sufficiently informative to assess sample utility. Experiment details are provided in **Section** B.1.

**Figure** 3 presents results across three datasets with varying data quality and complexity. PEAKS consistently outperforms random selection and other baselines, with one exception on CIFAR100 at the 30k data budget. However, the performance gains from data selection methods including

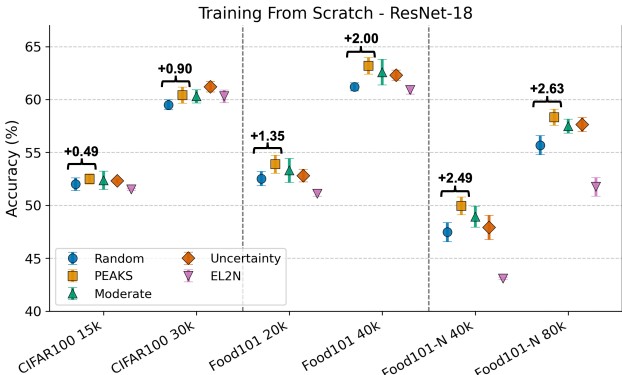

*Figure 3.* ResNet-18 without pretraining. Accuracy across three seeds. $|m|$: 7.5k (CIFAR100), 10k (Food101), 20k (Food101-N). WebVision experiments are omitted due to the computational cost of training from scratch.

PEAKS are modest, more so on the lower data size end and on the well-curated CIFAR100. This observation reveals once again that random selection serves as a surprisingly strong baseline for data selection on simpler datasets (Guo et al., 2022). Our result could also potentially be explained by the analytic theory of data pruning developed by Sorscher et al. (2022), which postulates that although in theory aggressive data pruning with high quality pruning metric can lead to bigger improvement compared to random pruning, in practice achieving such reliable metrics becomes challenging, especially with limited data. Furthermore, with a highly unreliable scoring function, beating random selection with small data budget might even be impossible.

This creates a fundamental tension: while aggressive pruning is desirable for selecting even more efficient data subset, stringent data budgets undermine pruning effectiveness, as training with less data would render the score unreliable. This paradox poses a limitation on all baseline methods that depend on model performance. We address this challenge by utilizing a pretrained model for our subsequent experiments with more aggressive pruning schemes, specifically a ViT-B/16 pretrained on ImageNet-1k using DINO (Caron et al., 2021) – see **Section** B.2 for details. The pretrained foundation provides a reliable feature space, enabling higher quality data selection decisions even with aggressive pruning rates.

This approach not only resolves the technical paradox but also aligns with contemporary practice, where pretrained models are increasingly ubiquitous and training from scratch is becoming less common. Thus, studying data selection in the context of fine-tuning with limited data offers both practical relevance and an ideal setting for understanding the full potential of data selection strategies.

## 5.2. Results With Pretraining

**With Validation Set: Table** 1 presents results where a portion of the test set is held out as validation data. This validation set enables us to establish performance upper bounds for data selection. Specifically, for Food101, Food101-N, and WebVision, the validation examples are human-curated, representing clear, artifact-free images with correct labels. We evaluate two upper bound baselines: Static, which trains on the entire validation set using epoch-based training, and Incremental, which performs IDS selecting only from the validation examples. These upper bounds are not computed for CIFAR100, as its test set is of similar quality to its training set. For WebVision, upper bounds are only available for the smallest data budget due to limited test set size. We utilize these validation sets to compute class prototypes for PEAKS-V and the three embedding-based methods.

The results show that PEAKS-V significantly outperforms all embedding baselines that are based on distance, demonstrating that such distance to class prototype alone is insufficient for data selection in the transfer learning setting. The integration of prediction error weighting in PEAKS provides substantial improvements. However, the performance gap between PEAKS and the upper bounds, particularly noticeable in Food-101N and WebVision, indicates room for improvement on complex real-world datasets. Moreover, we observe a considerable gap between static and incremental upper bounds in WebVision, which may indicate that the model's convergence on earlier selected samples hinders learning from newer examples. These findings suggest that IDS for complex datasets might benefit from more sophisticated regularization techniques or methods to improve the plasticity of gradient descent (Dohare et al., 2021).

**Without Validation Set:** Next, we evaluate our efficient PEAKS variant against baselines that do not require validation data. We also consider the best-performing embedding method (Moderate) from previous experiments, but use readout weights rather than validation set to measure distances.

Comparing **Tables** 1 and 2, we observe that PEAKS shows some performance degradation compared to PEAKS-V. This gap is most noticeable when samples per class are very limited (e.g., ×1), where output layer weights cannot accurately approximate class means. Despite this, PEAKS outperforms all baselines in most cases.

Notably, in the CIFAR100 experiments, the "wrong and low confidence" selection strategy outperforms PEAKS. This result can be attributed to the clean and well-curated nature of CIFAR100, where misclassifications often provide a reliable signal of sample difficulty. In such clean settings, low model confidence tends to correlate well with informativeness. However, this advantage diminishes on more complex datasets that contain label noise or outliers.

*Table 1.* Test accuracy using validation set. Results averaged across 5 seeds. ×1, ×2, and ×4 refer to dataset sizes of 2.5k, 5k, and 10k examples for CIFAR100, Food101, and Food101-N, and 25k, 50k, and 100k examples for WebVision, respectively.

| With Validation Set | CIFAR100 | Food101 | Food101-N | WebVision |
|---|---|---|---|---|
| Upper-Std (×1) | N/A | 55.6 (±2.1) | 55.6 (±2.1) | 59.5 (±0.7) |
| Upper-Inc (×1) | N/A | 55.2 (±1.1) | 55.2 (±1.1) | 50.6 (±1.0) |
| Random (×1) | 57.0 (±8.9) | 49.6 (±2.1) | 37.0 (±2.6) | 36.2 (±1.1) |
| Easy Emd (×1). | 48.6 (±7.7) | 44.3 (±1.7) | 39.2 (±1.6) | 33.6 (±3.4) |
| Moderate Emb. (×1) | 56.7 (±8.7) | 50.1 (±2.4) | 39.0 (±2.6) | 38.8 (±0.6) |
| Hard Emb. (×1) | 55.7 (±11.2) | 40.6 (±2.1) | 22.2 (±1.8) | 21.7 (±1.1) |
| PEAKS-V (×1) | **61.0** (±9.9) | **53.3** (±2.4) | **45.4** (±2.9) | **45.9** (±0.4) |
| Upper-Std (×2) | N/A | 66.5 (±1.0) | 66.5 (±1.0) | N/A |
| Upper-Inc (×2) | N/A | 63.8 (±2.1) | 63.8 (±2.0) | N/A |
| Random (×2) | 71.1 (±2.8) | 60.4 (±2.2) | 46.9 (±2.6) | 44.0 (±0.4) |
| Easy Emd. (×2) | 55.0 (±3.9) | 49.9 (±1.6) | 46.7 (±1.7) | 35.1 (±0.2) |
| Moderate Emb. (×2) | 68.9 (±2.4) | 61.9 (±2.0) | 50.5 (±2.2) | 47.5 (±0.3) |
| Hard Emb. (×2) | 71.8 (±2.5) | 49.1 (±2.7) | 20.6 (±1.5) | 18.7 (±0.3) |
| PEAKS-V (×2) | **75.5** (±2.1) | **64.2** (±2.0) | **56.1** (±2.6) | **54.4** (±0.3) |
| Upper-Std (×4) | N/A | 74.3 (±0.6) | 74.3 (±0.6) | N/A |
| Upper-Inc (×4) | N/A | 72.4 (±0.7) | 72.4 (±0.7) | N/A |
| Random (×4) | 81.1 (±0.5) | 69.7 (±1.8) | 55.0 (±2.6) | 50.0 (±0.3) |
| Easy Emd. (×4) | 65.5 (±1.1) | 56.4 (±1.4) | 52.9 (±1.6) | 38.9 (±1.1) |
| Moderate Emb. (×4) | 77.7 (±0.6) | 70.5 (±1.4) | 60.8 (±2.4) | 53.3 (±0.4) |
| Hard Emb. (×4) | 82.1 (±0.7) | 61.5 (±2.4) | 21.4 (±1.8) | 16.0 (±0.6) |
| PEAKS-V (×4) | **84.3** (±0.6) | **73.1** (±1.5) | **63.5** (±2.3) | **59.9** (±0.6) |

Furthermore, consistent with recent findings (Xia et al., 2023), selecting examples with intermediate distances to class means (Moderate Emb.) provides reliable baseline across all real-world datasets. Overall, while PEAKS achieves its best results with validation data, it maintains superior performance even in scenarios where validation data is unavailable, demonstrating its robustness and practical utility.

## 5.3. Impact of Selection Ratio

In previous experiments, we used a fixed selection rate of $p = 20\%$. This hyperparameter significantly influences the pace of the selection process for a fixed data budget $k$. Higher values ($p > 50\%$) make selection less discriminative by allowing lower-scored examples to be selected, leading to greater overlap with random selection. Conversely, lower values of $p$ focus strictly on the highest-scored samples, making selection more focused but potentially limiting diversity. If a selection method truly captures dataset characteristics and their relation to previously selected samples and the model state, we would expect it to perform better with lower $p$ and not benefit from the inclusion of low scored examples. We examine how varying this parameter affects performance across different methods on Food101 and Food101-N datasets, as shown in **Figure** 4.

The results show that none of the methods benefit from stricter selection ($p = 10\%$). In IDS, methods rely on the model state for selection, which is shaped by previously selected samples, but making optimal decisions incrementally without access to the entire data source makes it challenging to capture true data distributions.

*Table 2.* Test accuracies averaged across 3 seeds without validation set. Data budgets ×1, ×2 and ×4 are same with Table 1.

| Without Validation Set | CIFAR100 | Food101 | Food101-N | WebVision |
|---|---|---|---|---|
| Random (×1) | 58.0 (±5.9) | 48.9 (±1.8) | 37.2 (±3.0) | 36.6 (±0.6) |
| Moderate Emb. (×1) | 57.6 (±5.5) | 50.8 (±2.1) | 39.7 (±3.1) | 39.0 (±0.4) |
| EL2N (×1) | 60.1 (±6.7) | 46.2 (±2.1) | 32.6 (±2.4) | 32.2 (±2.0) |
| GraNd (×1) | **63.3** (±5.9) | 48.3 (±2.3) | 34.9 (±2.3) | 35.7 (±0.5) |
| Uncertainty (×1) | 62.0 (±6.0) | 49.1 (±1.8) | 36.3 (±3.3) | 35.2 (±0.3) |
| Wrong Low Conf. (×1) | 62.7 (±7.0) | 48.7 (±3.1) | 35.0 (±3.7) | 35.2 (±0.4) |
| PEAKS (×1) | 59.0 (±6.5) | **50.9** (±2.2) | **41.4** (±3.2) | **43.9** (±0.4) |
| Random (×2) | 70.2 (±3.3) | 59.6 (±2.0) | 46.5 (±3.0) | 44.1 (±0.3) |
| Moderate Emb. (×2) | 67.2 (±3.0) | 61.9 (±1.9) | 50.5 (±3.2) | 47.9 (±0.4) |
| EL2N (×2) | 74.2 (±3.6) | 56.6 (±2.3) | 38.0 (±2.4) | 37.5 (±0.2) |
| GraNd (×2) | 76.4 (±2.6) | 60.0 (±2.3) | 42.3 (±2.0) | 42.5 (±0.2) |
| Uncertainty (×2) | 75.9 (±3.2) | 61.1 (±2.4) | 45.7 (±3.3) | 40.6 (±0.4) |
| Wrong Low Conf. (×2) | **77.0** (±2.8) | 60.3 (±2.9) | 44.8 (±3.7) | 40.8 (±0.1) |
| PEAKS (×2) | 72.3 (±3.2) | **62.6** (±1.9) | **52.6** (±2.7) | **53.1** (±0.2) |
| Random (×4) | 79.4 (±1.3) | 69.2 (±1.9) | 55.8 (±3.1) | 50.1 (±0.1) |
| Moderate Emb. (×4) | 74.9 (±1.8) | 70.5 (±1.2) | 62.0 (±3.3) | 54.8 (±0.2) |
| EL2N (×4) | 82.5 (±0.9) | 66.5 (±2.1) | 44.7 (±3.4) | 41.5 (±0.6) |
| GraNd (×4) | 83.4 (±0.5) | 69.9 (±1.7) | 50.0 (±2.7) | 48.8 (±0.2) |
| Uncertainty (×4) | 82.6 (±0.6) | 71.5 (±1.8) | 55.7 (±3.0) | 45.6 (±0.3) |
| Wrong Low Conf. (×4) | **83.9** (±1.0) | 70.7 (±2.3) | 53.5 (±3.8) | 45.3 (±0.3) |
| PEAKS (×4) | 82.7 (±0.9) | **72.9** (±1.1) | **62.2** (±2.7) | **59.0** (±0.3) |

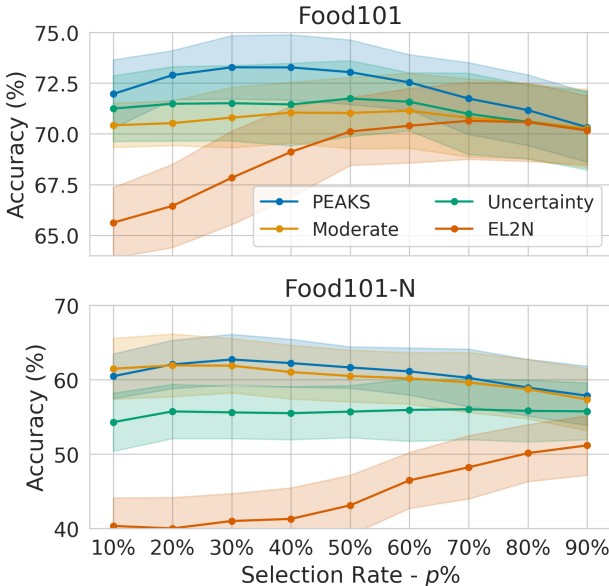

*Figure 4.* Performance comparison using a fixed budget of 10k samples from Food101 and Food101-N datasets with varying selection rates ($p\%$). Results averaged across three seeds.

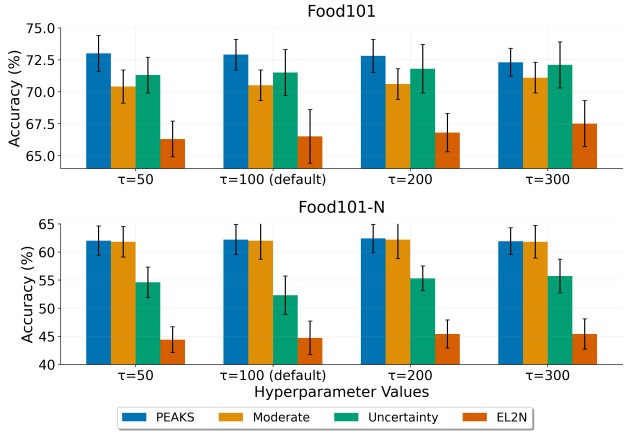

*Figure 5.* Impact of refresh period $\tau$ on performance. Results are averaged over three seeds, using a fixed selection budget of 10k examples from the Food101 and Food101-N datasets.

Moreover, since the scoring functions are approximations of the true utility, a larger selection fraction helps compensate for this uncertainty by accepting more samples that could be valuable. PEAKS maintains superior performance across most selection rates, achieving its peak performance at $p = 30\%$. As $p$ approaches $100\%$, all methods converge to similar accuracy levels, resembling random sampling. Notable, EL2N improves with higher selection rates. This can be explained by EL2N's focus on high-error examples when selection is strict, which may often correspond to noisy or atypical samples. As the selection rate increases, it naturally incorporates more diverse examples, leading to better performance.

### 5.4. Impact of Refresh Period ($\tau$)

Another hyperparameter in IDS is the refresh period $\tau$, which determines how frequently the score cache is reset during selection. This cache is used to compute percentile thresholds for dynamic selection criteria. While we do not expect selection quality to be highly sensitive to $\tau$, as it only affects the range of recent scores considered when computing percentiles, we nonetheless evaluate its impact empirically. We vary $\tau$ from 50 to 300 (default is 100 for all datasets except WebVision) on two datasets following the same configuration described in Section 5.3. As shown in Figure 5, we observe results are consistent across range of $\tau$ values.

### 5.5. Overlap in Selection

We next compare which samples are selected by different strategies. Since examples are sampled as $(x_p, y_p) \sim D_{\text{source}} \setminus T_t$, and $T_t$ varies across methods, the order in which samples appear differs, even with fixed random seeds. Fur-

thermore, randomness in the sampling process means not all samples are guaranteed to be encountered. To alleviate these variations, our analysis focuses on selecting 30% of samples from CIFAR-100 (15k) and Food-101 (∼23k). This larger data budget, compared to earlier experiments (**Section** 5.2), increases the likelihood of most samples being encountered at least once. Under these conditions, approximately 85% of all samples are seen by each method, with a high overlap (∼90%) in the samples encountered across methods (see **Section** B.3 for details). **Figure** 6 illustrates the Jaccard similarity between the final training sets $T_{\text{end}}$ selected.

PEAKS shows partial overlap (0.36–0.40) with EL2N, indicating that kernel similarity plays a significant role in guiding selection, too, rather than prediction error alone. Our analysis of Food101 samples selected exclusively by PEAKS (see **Section** D.3 in Appendix) suggests PEAKS may favor examples near decision boundaries between similar classes, while EL2N tends to select poorly framed or mislabeled samples. Interestingly, EL2N and PEAKS achieve the same accuracy (rounded to a single decimal point) on CIFAR100. Similarly, EL2N and Moderate exhibit comparable performance on Food101, despite having very low overlap in selection (0.18). These findings underscore an important aspect of data selection: multiple distinct subsets can serve as qualitatively equivalent training sets. Once a sample is selected and influences the model state, it inherently changes the utility of the remaining samples, leading to different but likely equally effective selection paths.

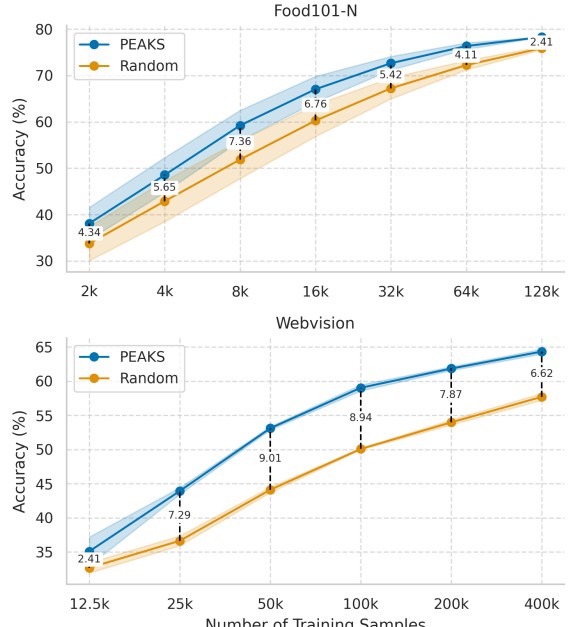

*Figure 7.* Accuracy vs. training dataset size for Food-101N and WebVision, averaged across 3 seeds (shaded areas show std.)

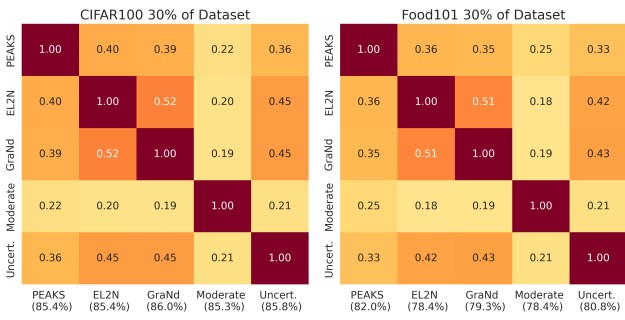

*Figure 6.* Jaccard similarity (intersection over union) between selected samples across different methods. Accuracies rounded to a single decimal point are shown in parentheses (single seed).

### 5.6. Scaling Against Random Selection

Finally, we evaluate PEAKS against random selection across a broader range of data budgets, focusing on Food101-N and WebVision due to their larger sizes. **Figure** 7 shows that the advantage of PEAKS over random selection becomes more significant as the dataset size increases. This is particularly evident in WebVision, where random selection requires approximately 4× more data to achieve similar performance levels. For instance, PEAKS with 50k and 100k samples achieves comparable accuracy to random selection using 200k and 400k samples, respectively. Similarly, on Food101-N, random selection consistently needs double the data to achieve comparable performance. These results demonstrate that effective data selection strategies can substantially reduce data requirements for DNNs, offering a practical path toward more efficient and sustainable training.

## 6. Conclusion

We introduced the Incremental Data Selection (IDS) problem, addressing data selection for DNNs in a dynamic con-

text where examples arrive as a continuous stream and must be selected without access to the full data source. IDS bridges the gap between the traditional coreset selection procedure and the practical needs of DNNs. Our analysis revealed fundamental relationships between newly arriving data samples and their potential for model improvement. These insights led to the development of PEAKS, an efficient data selection method tailored for IDS. Most notably, on WebVision, a large-scale real-world dataset, PEAKS achieves comparable performance while requiring only one-fourth of the data needed by random selection. Our results suggest that principled data selection strategies can substantially reduce data requirements for modern DNNs while maintaining model performance.

## Acknowledgements

Xingyu Zheng is supported by a William R. Miller Fellowship at the School of Biological Sciences at Cold Spring Harbor Laboratory. The authors are grateful to the ICML 2025 reviewers for their constructive comments.

## Impact Statement

This paper presents work whose goal is to advance the field of Machine Learning. There are many potential societal consequences of our work, none which we feel must be specifically highlighted here.

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

# A. IDS Algorithm

---

**Algorithm 1** Incremental Data Selection

---

1: **Input:** Initial set $T_0$, data budget $k$, selection rate $p$, refresh period $\tau$, Acquisition function $A(x_p, y_p, f_\theta; \mathcal{C})$
2: Train initial model $f_0$ on $T_0$
3: Initialize $\mathcal{C} \leftarrow \emptyset$, $t \leftarrow 0$, $S_{\text{new}} \leftarrow \emptyset$
4: **while** $|T_t| < k$ **do**
5:     $n_{\text{selected}} \leftarrow 0$
6:     **while** $n_{\text{selected}} < \delta$ and $|T_t| + n_{\text{selected}} < k$ **do**
7:         Sample $(x_p, y_p)$ from $\mathcal{D}_{\text{source}}$
8:         **if** $A(x_p, y_p, f_t; \mathcal{C})$ accepts **then**
9:             $S_{\text{new}} \leftarrow S_{\text{new}} \cup \{(x_p, y_p)\}$
10:             $n_{\text{selected}} \leftarrow n_{\text{selected}} + 1$
11:         **end if**
12:     **end while**
13:     batch $\leftarrow S_{\text{new}} \cup \text{Random}(T_t, b - \delta)$
14:     Update model $f_{t+1}$ using batch
15:     $t \leftarrow t + 1$
16:     $T_t \leftarrow T_{t-1} \cup S_{\text{new}}$
17:     $S_{\text{new}} \leftarrow \emptyset$
18:     **if** $t \bmod \tau = 0$ **then**
19:         $\mathcal{C} \leftarrow \emptyset$
20:     **end if**
21: **end while**
22: Fine-tune final model on $T_{\text{end}} = T_t$

---

# B. Experiment Details

## B.1. Details of Training from Scratch Experiments

We conducted our initial experiments using a ResNet-18 architecture trained from scratch on CIFAR100, Food101, and Food101-N datasets. For CIFAR100, following standard practice for low-resolution images, we modified the network architecture by replacing the initial 7×7 convolution layer (stride 2) and 3×3 max pooling layer (stride 2) with a single 3×3 convolution layer (stride 1).

**Training Configuration:**

- Optimizer: AdamW with learning rate 0.001 and weight decay 0.01

- Total training steps: 30000

- Batch size: 128

- Data normalization: Per-channel mean and standard deviation using ImageNet-1k statistics

- Data augmentation: Random cropping and random horizontal flipping ($p = 0.5$)

- Learning rate schedule: Reduced by factor of 10 at the start of final training phase

While training for 30000 steps allows models to overfit the training dataset, we did not observe any adverse effects on validation performance. We picked the optimizer, learning rate, and weight decay based on initial phase training performance (same for all baselines, with randomly selected $m$ examples).

**Dataset Configurations:**

- CIFAR-100: 15k and 30k samples

- Food-101: 20k and 40k samples

- Food-101N: 40k and 80k samples

We selected these dataset sizes to represent two distinct scenarios: a minimal viable budget that enables successful training from scratch (first number), and a doubled budget to study larger budget. In the literature, dataset size is commonly reported as a fraction of the total dataset. However, our datasets vary drastically in terms of total samples. For example, 30% of CIFAR100 means 15000 samples, while in Food101-N the same ratio means 90000 samples. Therefore, we focus on absolute sample counts rather than fractions. Due to computational constraints, we could not conduct exhaustive experiments across a wider range of dataset sizes. Training from scratch is particularly resource-intensive compared to fine-tuning pre-trained models, which influenced our decision to focus on these specific dataset sizes. For similar computational reasons, we excluded WebVision from our training-from-scratch experiments.

**IDS-Specific Parameters:**

- Selection increment ($\delta$):
    - 4 for CIFAR-100 and Food-101
    - 8 for Food-101N (adjusted for larger dataset size)

- Cache Refresh Period ($\tau$): 400 steps

- Selection Rate ($p$): 20%

- Initial Dataset Size ($m$): CIFAR-100 7.5k, Food-101 10k, Food-101N: 20k. In other words, half of the low budget scenario.

- Initial Training Steps: 10000

The cache refresh period $\tau$ proved robust across a wide range of values. However, if $\tau$ is set too high or cache refreshing is not performed frequently enough, selection rates consistently drop as it becomes harder to collect new samples. This occurs because selection scores (PEAKS, Moderate, EL2N, and Uncertainty) typically decrease as the model learns, and without regular cache refreshes, high scores from early training prevent newer examples from being selected.

### B.2. Details of Pretraining Experiments

Our primary focus was on data selection for fine-tuning pretrained models, reflecting their ubiquity in practice as training from scratch becomes increasingly uncommon. For these experiments, we used a ViT-B/16 architecture pretrained using DINO self-supervised learning on ImageNet-1k.

**Training Configuration:**

- Optimizer:
    - WebVision: AdamW (lr=0.0001, weight decay=0.01)
    - All other datasets: SGD with momentum (lr=0.001, weight decay=0.0001)

- Training steps:
    - WebVision: 6000
    - All other datasets: 2000

- Batch size: 128

- Data normalization: ImageNet-1k per-channel mean and standard deviation. We upscaled CIFAR100 images to $224 \times 224 \times 3$.

- Data augmentation: None

- Learning rate schedule: None

Notably, we omitted data augmentation and learning rate scheduling, which are less critical when fine-tuning pretrained models compared to training from scratch. This simplification also enables clearer analysis of the relationship between dataset size and model performance. Similar to the training-from-scratch case, we picked the optimizer, learning rate, and weight decay based on initial phase training performance.

**Dataset Configurations:**

- CIFAR-100, Food-101, Food-101N: 2.5k, 5k, 10k samples

- WebVision: 25k, 50k, 100k samples

These sizes correspond to approximately 25, 50, and 100 samples per class in the training dataset.

**IDS-Specific Parameters:**

- Selection increment ($\delta$):

    - CIFAR-100, Food-101, Food-101N: 2, 4, and 8 for respective data budgets
    - WebVision: 5, 14, and 30.
    - These values ensure approximately half of the training steps occur during the data selection phase.

- Cache Refresh period ($\tau$):

    - WebVision: 300
    - Other datasets: 100
    - Configured for roughly 10 refreshes during selection

- Selection rate ($p$): 20%

- Initial dataset size ($m$):

    - WebVision: 10000
    - Other datasets: 1000
    - Approximately 10 samples × number of classes, though not necessarily class-balanced

- Initial training steps:

    - WebVision: 1000
    - Other datasets: 100

These hyperparameters were intentionally set to round values based on dataset characteristics without extensive optimization. This approach avoids potential bias, as different selection methods may achieve optimal performance under different configurations. Additionally, it provides a more realistic evaluation, as optimal parameter tuning may be impractical in real-world scenarios where data sources can be unbounded.

The above configurations were consistent across all experiments with or without validation set. For experiments involving validation sets, we made the following additional considerations:

**Validation Set Configuration:**

- Food-101 and Food-101N: Reserved clean, human-annotated samples from test set matching training budget (2.5k, 5k, 10k)

- WebVision: Fixed 25k validation set due to limited test set size (50k total)

- CIFAR-100: Reserved 2.5k, 5k, 10k from training set due to test set size limitations (10k) and similar distribution to training data.

For validation set experiments, we used 5 random seeds (versus 3 for other experiments) to obtain more reliable generalization estimates given the reduced test set size. The validation sets were used to compute class mean embeddings every $\tau$ steps for embedding-based methods (Table 1).

## B.3. Details of Overlap in Selection

In **Section** 5.5, we analyze the overlap between samples selected by different methods using Jaccard Similarity (intersection over union). It is important to note that Jaccard Similarity operates on sets and is therefore insensitive to ordering. For instance, if two methods both select the same sample but assign it different importance rankings, the Jaccard Similarity treats these equally as selections.

Our experimental setup follows the configurations detailed in **Section** B.2 for pretrained models, with one key modification: we increased the selection increment $\delta$ to 16 to ensure sufficient examples could be selected within the allocated training steps. The initial training dataset of size $m = 1000$ was excluded from Jaccard Similarity calculations as these samples were randomly collected rather than chosen by selection methods.

To validate that methods encountered similar samples during the selection phase, we also computed the Jaccard Similarity of seen samples (see **Figure** 8). This analysis confirmed that approximately 90% of samples seen by one method were also encountered by others, ensuring a fair comparison of selection strategies despite the sequential nature of IDS.

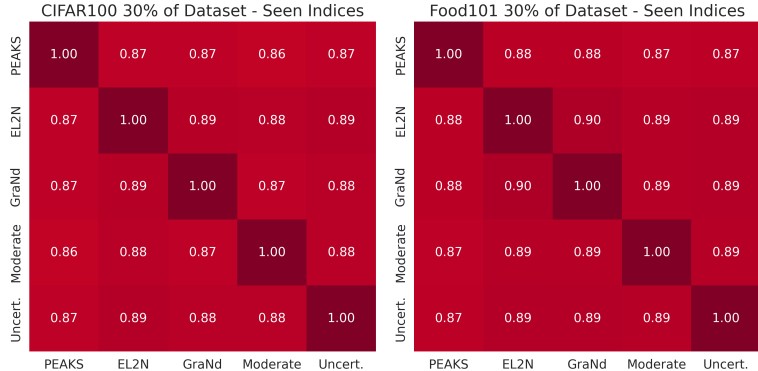

*Figure 8.* Jaccard similarity between sets of samples encountered (seen) by different selection methods during training on CIFAR-100 (left) and Food-101 (right) when selecting 30% of the dataset. High values ( 0.87-0.90) across all method pairs indicate that methods encounter largely overlapping sets of samples during selection, despite making different selection decisions.

## B.4. Details of Scaling Against Random Selection Experiments

For the scaling experiments presented in **Section** 5.6, we followed the experimental configuration described in **Section** B.2, adjusting only the selection increment $\delta$ based on dataset size to ensure balanced data collection throughout the training process.

For Food-101N, we started with $\delta = 1$ for the 2k budget and doubled it for each subsequent budget doubling. For WebVision:

- 12.5k budget: $\delta = 5$

- 25k-100k budgets: Same configuration as **Section** B.2

- 200k budget: $\delta = 60$

- 400k budget: $\delta = 100$

These adjustments ensure that data selection keeps pace with model training, avoiding both too rapid and too slow data collection.

## B.5. Choice of Datasets

Our experiments require datasets that enable a thorough assessment of IDS across varying data quality conditions while maintaining reliable evaluation metrics. We selected four datasets—CIFAR100, Food101, Food101-N, and WebVision—based on several criteria that align with our experimental objectives (see **Table** 3 for dataset statistics).

| Dataset | Num Classes | Train Size | Test Size | Train Label Balance | Train Data Noise |
|---|---|---|---|---|---|
| CIFAR100 | 100 | 50k | 10k | Yes | ∼0% |
| Food101 | 101 | 75k | 25k | Yes | ∼8% |
| Food101-N | 101 | 310k | 25k | Mild Imbalance | ∼20% |
| WebVision | 1000 | 2.4M | 50k | Strong Imbalance | ∼20% |

*Table 3.* Dataset statistics and characteristics used in our experiments.

The primary consideration in our dataset selection was the need to evaluate IDS across a spectrum of data quality scenarios. CIFAR100 serves as our clean, well-curated baseline dataset, being both balanced and widely accepted in data selection literature. Food101 represents a middle ground, containing approximately 8% label noise primarily manifesting as color intensity variations and occasional labeling errors. Food101-N introduces more substantial quality variation with an estimated 20% noise level, while WebVision provides a large-scale real-world scenario with naturally occurring noise patterns.

Several crucial factors influenced our dataset selection: First, reliable evaluation necessitates clean and sufficiently large test sets, particularly as we partition these for validation in PEAKS-V experiments and upper bound analysis. Both Food101-N and WebVision satisfy this requirement by providing human-verified test sets of relatively large size.

Second, we explicitly avoided datasets closely related to ImageNet (such as Tiny ImageNet) to prevent potential bias, as ImageNet serves as our pretraining source.

Third, the datasets needed to contain sufficient training examples to make the selection process meaningful. We excluded several otherwise suitable datasets that contained not enough examples (e.g., less than 50k).

Finally, we prioritized well-established datasets to ensure reproducibility and contextual relevance. While Food101 and WebVision are less commonly used in coreset selection literature, they are widely recognized in broader computer vision and machine learning research, particularly in studies involving learning from noisy data and real-world distributions.

### B.6. Choice of Baselines

We compare PEAKS against eight baseline methods that can be naturally adapted to the IDS setting. Our selection prioritizes methods that can make independent decisions for each example without requiring access to a pool of candidates. The baselines can be categorized into four groups based on their selection criteria:

**Embedding-based Methods.** We implement three variants that utilize penultimate layer embeddings:

- **Easy Embedding:** Selects examples with the highest cosine similarity to their class embedding (also known as "Herding" in literature). The class embedding is computed either via validation set means or output layer weights, depending on the context.

- **Hard Embedding:** Selects examples with the lowest cosine similarity to their class embedding, targeting the least prototypical examples.

- **Moderate Embedding:** Selects examples with intermediate similarity scores. For a selection rate of $p\%$, it chooses examples whose distances fall between the $(50 - \frac{p}{2})$th and $(50 + \frac{p}{2})$th percentiles in cache $\mathcal{C}$.

**Error-based Method.** We include EL2N, which selects examples based on the $L_2$ norm of the error vector (difference between predicted probabilities and one-hot encoded labels).

**Gradient-based Method.** GraNd selects examples with larger gradient norm magnitudes. This method is computationally most expensive as it requires computing per-example gradients through backpropagation.

**Uncertainty-based Method.** We implement the "least confidence" metric that selects examples based on $1 - \max(\text{softmax}(\text{logits}))$. This is uniquely unsupervised, requiring no label information.

**Wrong and Low Confidence Method.** Wrongly predicted samples but with lower confidence. We assign a score of 0 to correctly predicted examples and a score of 1 - max(softmax) to incorrect ones.

**Random Selection.** This is uniform random sampling from $D_{source}$.

Several popular data selection methods were not included in our evaluation due to incompatibility with the IDS setting:

- Methods requiring a pool of examples to approximate dataset-wide characteristics

- Approaches based on bilevel optimization or multiple model training

- Methods requiring auxiliary reference models

These exclusions stem from either computational constraints or fundamental incompatibility with the sequential nature of IDS. Rather than potentially misrepresenting these methods by modifying them beyond their intended use, we focus on baselines that naturally adapt to the incremental setting while maintaining reasonable computational requirements.

### B.7. IDS Practical Considerations

When implementing IDS at scale, several practical considerations become important. First, processing individual examples through forward passes is computationally inefficient, as it fails to leverage the parallelization capabilities of modern deep learning frameworks and hardware. To address this, we process potential examples in batches during the selection phase. While we compute model outputs for the entire batch in parallel, the selection process remains sequential: we evaluate examples in order from the start of the batch using our acquisition function, select the first $\delta$ examples that meet our criteria, and discard the remaining examples. This approach maintains the incremental nature of our selection process and does not utilize any information across samples in the batch while allowing us to benefit from batched forward passes for computational efficiency.

Second, instead of updating the training set $T_i$ immediately after each selection, we maintain a buffer of recently selected examples and integrate them into $T_i$ during the cache $\mathcal{C}$ refresh operation every $\tau$ updates. This deferred update strategy aligns better with typical data loader implementations, which often prefetch data for future updates. Our empirical evaluation shows no significant performance degradation compared to immediate updates, while the implementation becomes more efficient and robust.

Finally, we sample potential examples from $D_{source} \setminus T_i$ rather than directly from $D_{source}$. This ensures that already selected examples are not reconsidered, eliminating the need for explicit duplicate detection. We do not require any method to detect exact duplicates as in practice this can be handled perfectly with simple data structures.

These tricks are consistently applied across all baseline methods in our experiments, ensuring fair comparisons. These practical considerations enable efficient scaling of IDS while preserving its core theoretical properties and selection behavior.

## C. Extended Related Work

**Active Learning:** In Active Learning (AL), a model queries an oracle to label the most informative examples from a large unlabeled pool (Ren et al., 2021; Settles, 2009; Huang et al., 2018). Streaming AL scenarios (Cacciarelli & Kulahci, 2024), which evaluate examples incrementally as they arrive, are most related to our work. However, key differences exist. AL aims to minimize labeling costs, while IDS assumes labels are readily available, simplifying sample evaluation.

However, IDS introduces unique challenges. In AL, decisions about labeling a sample can often be deferred until more information is gathered. In contrast, IDS requires immediate decisions. Also, while AL typically emphasizes minimizing labeling costs and often involves resetting and retraining the model on the final dataset, IDS simultaneously trains the model efficiently during the data selection process. Thus, while AL and IDS have complementary objectives, they operate under distinct goals and constraints.

**Robust, Long-Tailed, and Continual Learning:** In IDS, the data source $D_{\text{source}}$ may be noisy and class-imbalanced. Robust learning with noisy labels has been extensively studied (Song et al., 2022; Zhou, 2018). Similarly, Long-Tailed Learning addresses scenarios where a subset of classes has disproportionately many samples, while most classes are underrepresented (Zhang et al., 2023). Although these challenges are relevant to IDS, they are not its primary focus.

Continual Learning (CL) methods aim to train DNNs on evolving data distributions while retaining previously acquired knowledge (Wang et al., 2023; Delange et al., 2021). A common strategy in CL is to replay past examples by mixing them with new data during training—an approach that bears resemblance to the model update mechanism in IDS, where previously selected samples are combined with new selections. This similarity, along with IDS's inherently incremental setup, suggests a potential connection between the two paradigms.

However, our focus in this work is solely on data efficiency under the assumption that $D_{\text{source}}$ remains stationary. It is important to note that PEAKS is explicitly designed to select samples that maximize changes in model logits—an intuitive objective for standard learning scenarios. In the context of CL, however, such samples may not be optimal for mitigating catastrophic forgetting or promoting knowledge retention and transfer—key objectives in CL. Addressing these challenges would likely require alternative selection criteria. Extending IDS and PEAKS to the CL setting is a promising direction for future work, but one that would necessitate significant conceptual and algorithmic adjustments.

## D. Additional Results

### D.1. Approximate vs. Exact PEAKS Scores.

In Section 3, we introduced an exact score for evaluating example utility (Eq. (3)), derived directly from first-order approximations of model updates. However, computing Eq. (3) requires the Jacobian of every validation sample with respect to all model parameters—a matrix of size (number of parameters) × (number of classes)—making it computationally prohibitive for modern networks.

To address this, we made two simplifications. First, we assumed that most learning occurs in the final classification layer, allowing us to approximate the effect of a sample using only the final layer weights (Eq. (9)). This is supported by prior work showing that (i) early layers are more transferable across tasks (Yosinski et al., 2014; Neyshabur et al., 2020), (ii) pretraining leads shallow layers to converge more quickly (Chen et al., 2023), and (iii) early layers change less during fine-tuning (Goerttler & Obermayer, 2024).

Second, to avoid the need for a validation set, we approximated class mean embeddings using the output weights of the final layer, yielding Eq. (11). This formulation underlies the main variant of PEAKS used in our experiments.

To empirically assess the quality of these approximations, we compare all three scoring variants—Eq. (3), Eq. (9), and Eq. (11)—in a controlled setup using CIFAR10 and CIFAR100. Due to computational constraints, we use ResNet-18 instead of ViT.

We begin by pretraining a ResNet-18 on CIFAR100, then fine-tune it on 3000 CIFAR10 examples to simulate the initialization phase used in IDS. We hold out 500 additional CIFAR10 samples for validation and evaluate the selection scores of 1000 unseen examples using all three scoring variants.

Figure 9 presents our comparison. On the left, we report Spearman rank correlations between scoring methods. The center and right plots show pairwise scatter plots of example rankings under exact vs. approximate methods. We observe strong agreement in ranking between the exact and approximate scores, indicating that our simplifications preserve the key ordering needed for effective selection.

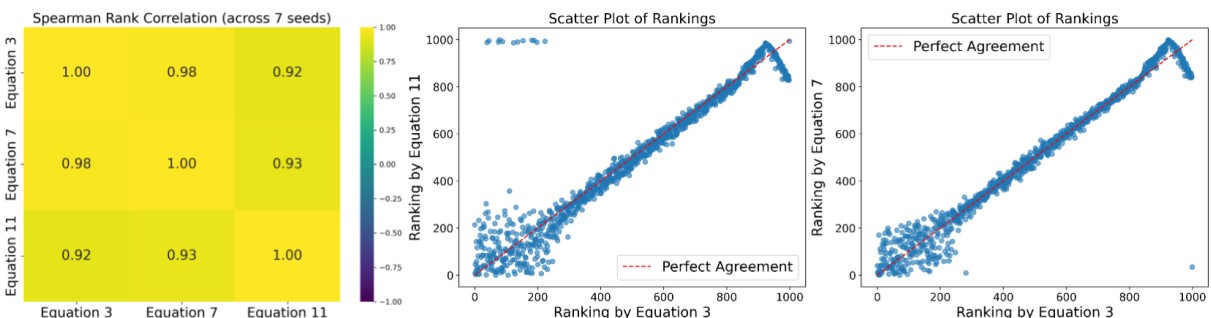

*Figure 9.* Left: Spearman rank correlation of 1000 examples scored by exact and approximate methods. Middle: Score ranking agreement between Eq. (3) and Eq. (9). Right: Agreement between Eq. (3) and Eq. (11). Scatter plots are based on a single seed.

## D.2. Why Do We Need Cache $\mathcal{C}$ Refresh?

To empirically demonstrate the necessity of cache refreshing discussed in **Section** 4.2, we analyze how selection scores evolve during training. We focus on selecting 30% of samples from CIFAR-100 (15k) and Food-101 ( 23k), consistent with our overlap analysis setting in **Section** 5.5. Since different methods operate on vastly different scales (e.g., Moderate selection's cosine distance is bounded while GraNd can be arbitrarily large), we normalize each score by its maximum value of rolling average for comparison.

**Figure** 10 illustrates the evolution of these normalized scores across training. Several key patterns emerge. All methods show a general decreasing trend in their scores as the model learns, though at different rates. PEAKS exhibits the sharpest decline, reflecting its strong dependence on prediction error which naturally decreases during training. Moderate selection maintains relatively stable scores compared to other methods, though still showing a gradual decrease. This declining pattern highlights why cache refreshing is crucial: without periodic resets, early high scores would dominate the cache's high percentile band, making it increasingly difficult to select new samples. While our chosen $\tau$ works well in practice, these observations suggest that optimal refresh rates could potentially be method-specific, given their different temporal dynamics.

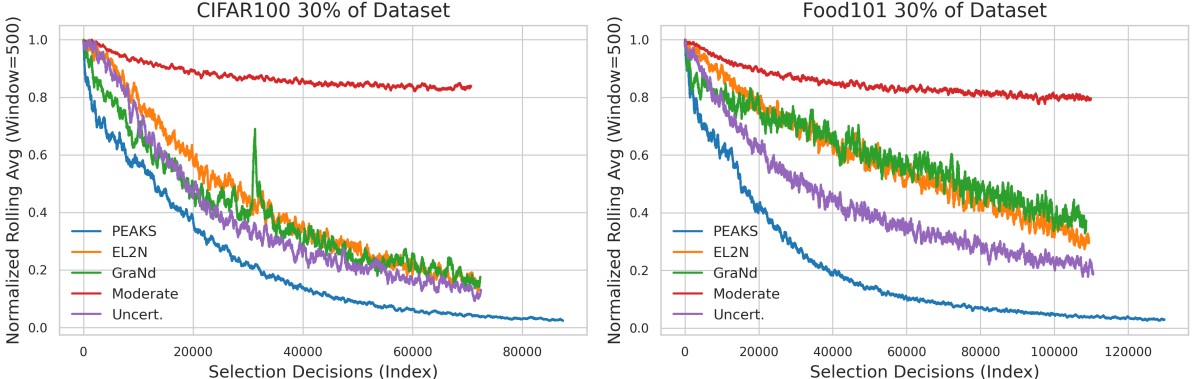

*Figure 10.* Evolution of normalized selection scores across training for different methods. Each line shows a rolling average (window size=500) of selection scores, scaled relative to their maximum rolling mean to bound scores between 0 and 1. This normalization enables comparison of temporal trends across methods despite their different scales.

## D.3. What Does PEAKS Capture Additionally Compared to EL2N?

To better understand how PEAKS' kernel similarity term influences selection decisions compared to pure error-based methods like EL2N, we conducted a detailed analysis of their selections on Food-101. We chose this dataset for its visual interpretability, as opposed to the low-resolution CIFAR-100 or the complex 1000-class WebVision dataset.

We analyzed selections during the middle 100 steps of the data selection phase when selecting 30% of Food-101 ($\sim$23k samples), consistent with our setting in **Section** 5.5. During this period, both methods encountered approximately 4000 examples. Due to the sequential nature of IDS, we focused on the 167 examples seen by both methods. Among these, 23 were exclusively selected by PEAKS and rejected by EL2N, while 24 showed the opposite pattern. **Figure** 11 visualizes the top 10 highest-scoring examples from each group, annotated with ground-truth labels and top predicted classes.

PEAKS' exclusive selections reveal an interesting pattern: they often represent examples that lie at the boundary between visually similar classes. For instance, the model shows uncertainty between spaghetti carbonara and bolognese (image-1), club sandwich and grilled cheese sandwich (image-2), or poutine and french fries (image-3). While this observation is somewhat anecdotal, it suggests that PEAKS identifies examples that could help establish more robust decision boundaries between similar classes. In contrast, EL2N's exclusive selections tend to favor examples that are poorly framed, potentially mislabeled, or outliers. This difference highlights a limitation of using prediction error alone as a selection criterion for real-world datasets, where high error might indicate problematic samples rather than informative examples.

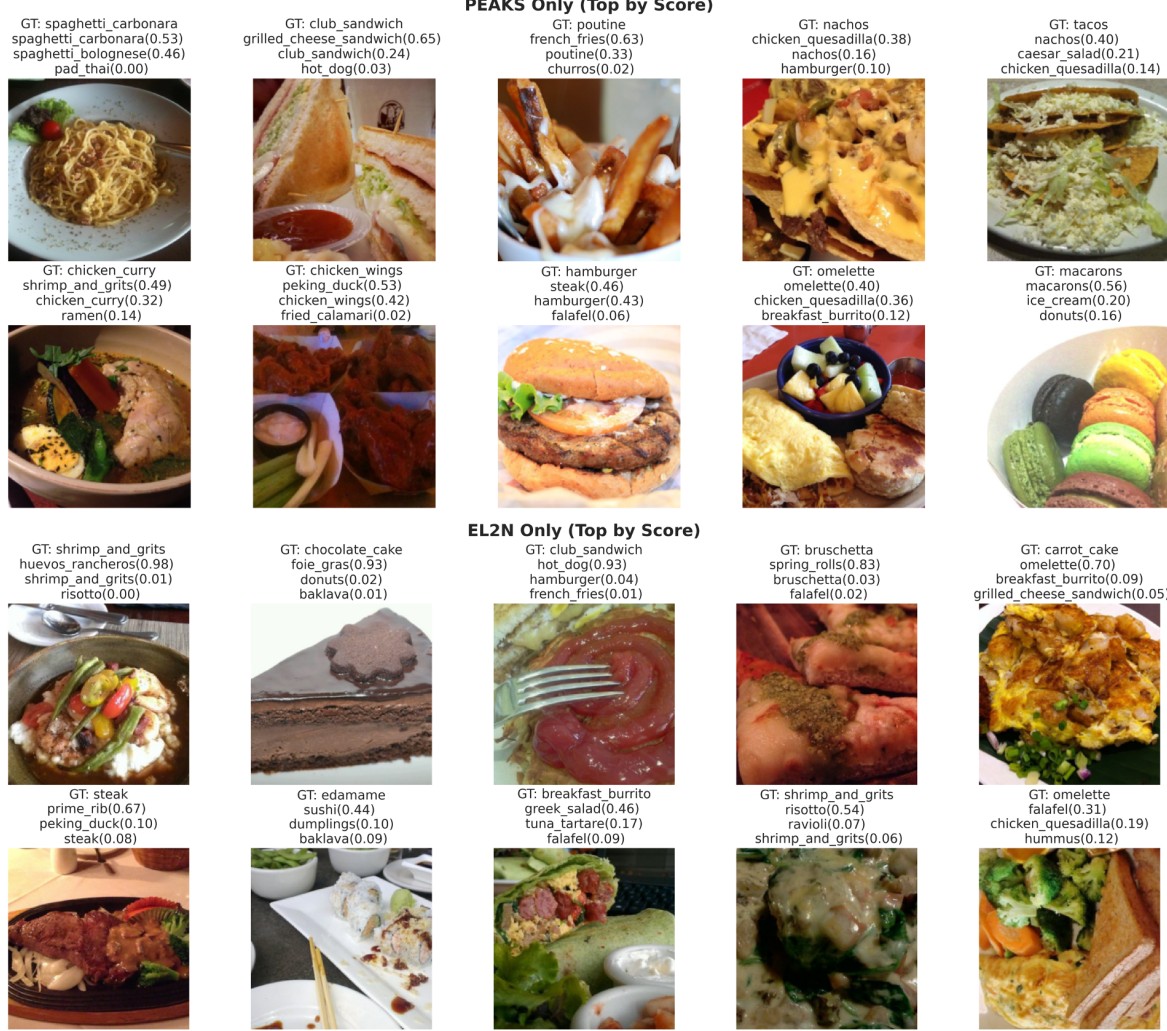

*Figure 11.* Shown above are two separate 2×5 grids of images from the Food101 dataset encountered in the middle of the data selection phase. The first figure highlights the top 10 images (by score) that were exclusively selected by PEAKS but not by EL2N, while the second figure shows the top 10 images (by score) exclusively selected by EL2N but not by PEAKS. Each image is annotated with its ground-truth label and the top three predicted classes (with their respective softmax probabilities). This comparison illustrates the differences in selection criteria between the two methods.

### D.4. Alternative Sampling Mechanism

During the data selection phase of IDS, each training batch combines $\delta$ newly selected examples with $b - \delta$ randomly sampled examples from the existing training set, where $b$ is the batch size. However, uniform random sampling of previous examples leads to an uneven usage distribution: some early-selected samples are repeatedly chosen while others remain underutilized. **Figure** 12 illustrates this phenomenon, showing significant variation in sample usage counts. While our final fine-tuning phase ensures the model eventually learns from all selected examples, the uneven sample distribution during the selection phase may bias the model's learning. To address this, we implement a simple count-based sampling strategy. For each example $i$, we track its usage count $c_i(t)$ and assign sampling probabilities inversely proportional to these counts:

$$p_i(t) = \frac{1/c_i(t)}{\sum_{j=1}^{n} 1/c_j(t)} \tag{14}$$

This approach reduces the probability of selecting frequently used samples while promoting the use of underutilized ones. As shown in **Table** 4, this simple modification improves accuracy in 8 out of 9 dataset and budget combinations tested with

*Table 4.* Alternative sampling using counts. Test accuracies averaged across 3 seeds without validation set. Data budgets ×1, ×2, and ×4 refer to dataset sizes of 2.5k, 5k, and 10k examples

| Without Validation Set | CIFAR100 | Food101 | Food101-N |
|---|---|---|---|
| PEAKS Alt. Sampling (×1) | 60.0 | 51.5 | 42.0 |
| PEAKS (×1) | 59.0 | 50.9 | 41.4 |
| PEAKS Alt. Sampling (×2) | 73.8 | 63.0 | 52.8 |
| PEAKS (×2) | 72.3 | 62.6 | 52.6 |
| PEAKS Alt. Sampling (×4) | 83.5 | 73.1 | 61.9 |
| PEAKS (×4) | 82.7 | 72.9 | 62.2 |

PEAKS. These results suggest that the batch formation strategy plays an important role in IDS Future work could explore more sophisticated batch formation functions $\mathcal{B}(T_i, f_{\theta_t}; \theta_B)$, where $\theta_B$ represents adaptive sampling parameters that evolve with the model state and training dynamics.

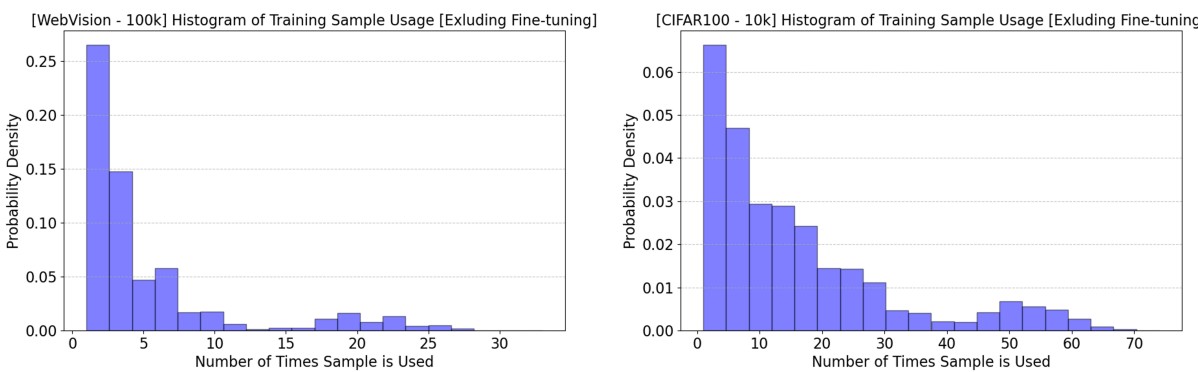

*Figure 12.* Sample repetition during training (excluding fine-tuning phase) for WebVision and CIFAR100 under random sampling.

### D.5. Ablation Study - Class Normalization Term for PEAKS

In PEAKS, we normalize the expected improvement score $\mathbb{E}_{D_v^{y_p}}[\Delta(x_p, x_v)]$ by $\frac{1}{c_{y_p}(t)}$, where $c_{y_p}(t)$ represents the number of samples already selected from class $y_p$ at time $t$. This normalization accounts for varying score ranges across different classes. **Table** 5 shows ablation results across three datasets (WebVision omitted due to computational constraints). The results demonstrate that this term consistently improves performance of PEAK on both balanced and imbalanced datasets.

*Table 5.* Impact of class normalization for PEAKS. Test accuracies averaged across 3 seeds without validation set. Data budgets $\times 1$, $\times 2$, and $\times 4$ refer to dataset sizes of 2.5k, 5k, and 10k examples

| Without Validation Set | CIFAR100 | Food101 | Food101-N |
|---|---|---|---|
| PEAKS w/o $c_{y_p}$ ($\times 1$) | 57.8 | 49.8 | 39.2 |
| PEAKS ($\times 1$) | 59.0 | 50.9 | 41.4 |
| PEAKS w/o $c_{y_p}$ ($\times 2$) | 71.8 | 61.7 | 50.5 |
| PEAKS ($\times 2$) | 72.3 | 62.6 | 52.6 |
| PEAKS ($\times 4$) | 82.7 | 72.3 | 59.6 |
| PEAKS ($\times 4$) | 82.7 | 72.9 | 62.2 |

