# OpenReview forum: "PEAKS: Selecting Key Training Examples Incrementally via Prediction Error Anchored by Kernel Similarity"
_ICML.cc/2025/Conference — ICML 2025 poster_

### Official Review · Reviewer_dKNe · 2025-03-11

**Overall Recommendation:** 2

**Summary:**

The paper introduces an algorithm for Incremental Data Selection (IDS) that selects training examples from a continuous data stream by combining prediction error and kernel similarity. The problem is important for the machine learning community. IDS addresses the challenge of efficient data utilization in deep learning, particularly in scenarios where full datasets are unavailable upfront. PEAKS dynamically balances samples with high prediction errors (indicating model uncertainty) and those aligned with class prototypes (via kernel similarity). Experiments on image datasets (CIFAR100, Food101, WebVision) demonstrate PEAKS' superiority over baselines like EL2N and GraNd, especially in low-data regimes.

**Claims And Evidence:**

Yes

**Essential References Not Discussed:**

[1] Cost-Effective training of deep CNNs with active model adaptation. KDD 2018

See weaknesses.

**Experimental Designs Or Analyses:**

Yes

**Methods And Evaluation Criteria:**

Yes

**Other Comments Or Suggestions:**

NA

**Other Strengths And Weaknesses:**

# Strengths
1.	The problem of IDS is important for the machine learning community.
2.	The proposed method employs a measure that considers two core factors.
3.	The proposed method is evaluated on several vision datasets against other baselines.
# Weaknesses
1.	The core idea of PEAKS combines prediction error and kernel similarity. While the integration is practical, the theoretical contribution is incremental rather than groundbreaking, lacking a novel framework. Besides, the proposed method shares concepts in the active learning field [1], while proper discussions are not included in this paper.
2.	The experiments are restricted to image classification tasks. There is no validation on non-vision domains (e.g., NLP, time-series) or regression tasks, raising concerns about generalizability.
3.	While PEAKS is framed as efficient, computational overhead from kernel similarity calculations, cache maintenance, and dynamic thresholds is not quantified. This omission leaves scalability for large-scale models/datasets unclear.
4.	PEAKS-V relies on a validation set for class prototypes, which is often impractical in streaming scenarios. The validation-free PEAKS variant underperforms other baselines, suggesting robustness issues in real-world deployments. Besides, the authors only report the average accuracy while ignoring the variance in the Tables, making it hard to understand the robustness.
5.	The paper does not address how sensitive PEAKS is to the choices on hyperparameters like selection rate and refresh period, limiting reproducibility and adaptability to diverse settings.
6.	When using pre-trained models as the initialization, another concern also emerges, i.e., the is overlapping between the pre-training dataset and the downstream tasks. If the pre-trained model has already known them, why bother conducting further learning processes?

[1] Cost-Effective training of deep CNNs with active model adaptation. KDD 2018

**Questions For Authors:**

NA

**Relation To Broader Scientific Literature:**

The paper introduces an algorithm for Incremental Data Selection (IDS) that selects training examples from a continuous data stream by combining prediction error and kernel similarity. The problem is important for the machine learning community. IDS addresses the challenge of efficient data utilization in deep learning, particularly in scenarios where full datasets are unavailable upfront. PEAKS dynamically balances samples with high prediction errors (indicating model uncertainty) and those aligned with class prototypes (via kernel similarity). Experiments on image datasets (CIFAR100, Food101, WebVision) demonstrate PEAKS' superiority over baselines like EL2N and GraNd, especially in low-data regimes.

**Theoretical Claims:**

Yes

---

> ### Author Rebuttal · Authors · 2025-03-31
>
> We are thankful to the reviewer for their detailed assessment and helpful suggestions.
>
> **Weakness-1**
>
> We acknowledge that our theoretical contribution builds incrementally on existing frameworks rather than proposing an entirely new theoretical foundation. Our primary contributions are twofold. First, we formulate the Incremental Data Selection (IDS) problem — a practical, previously unexplored setting that addresses real-world constraints in data selection. Second, we devise the PEAKS algorithm tailored for the IDS.
>
> We also thank the reviewer for highlighting the connection to AL. We currently have a brief discussion of AL in Appendix C and will move this to the main paper, expanding it to include the suggested reference.
>
> **Weakness-2**
>
> We selected image classification as a testbed for IDS as it is a prevalent and well-established task in the data selection literature (e.g., [1, 2]). While PEAKS' design is not specific to the image domain, we acknowledge that the absence of empirical evaluation on other tasks is a limitation. We commit to adding a discussion of this limitation in the revised paper.
>
> [1] Beyond neural scaling laws: beating power law scaling via data pruning, NeurIPS 2022.
>
> [2] Deep Learning on a Data Diet: Finding Important Examples Early in Training, NeurIPS 2021.
>
>
> **Weakness-3**
>
> We thank the reviewer for raising this point and will clarify these details in the paper. Due to the approximations in Section 3, kernel similarity in PEAKS reduces to the product of the error and output logit (Eq.11), incurring no extra cost. Regarding cache and thresholding (which are not specific to PEAKS but inherent to the IDS), the required overhead is minimal.
> 1. The cache stores only $\tau \times \delta$ floating-point values. For example, in the Section 5.2 experiments, the smallest cache stores 200 values (0.78 KB), while the largest (WebVision, 100k budget) stores 9000 values (35 KB).
> 2. Dynamic thresholding requires maintaining a sorted cache for percentile computation. These overheads are negligible compared to the training cost.
>
> **Weakness-4 (performance of PEAKS)**
>
> We would like to respectfully clarify that the validation-free PEAKS variant consistently demonstrates strong performance across experiments. As demonstrated in Figure 3, Table 2, and Figure 4, the validation-free PEAKS consistently outperforms all baselines across datasets and budgets, performing slightly worse only on the simpler CIFAR100 dataset. Notably, on the challenging real-world WebVision dataset, PEAKS significantly outperforms the closest baselines by margins of 4.9%, 5.2%, and 4.2% across the three evaluated data budgets. This demonstrates that PEAKS is the most robust among baselines for real-world scenarios.
>
> **Weakness-4 (accuracy variance)**
>
> Thank you for this important point about reporting statistical variance. Due to space limitations in this response, we cannot present tables here. However, we can confirm that the accuracy variance is within an acceptable range compared to the performance differences between baselines. The complete variance data for all experiments will be included in the revised manuscript.
>
> **Weakness-5**
>
> We address selection rate sensitivity in Section 5.3, where Figure 4 shows PEAKS consistently outperforms other methods across selection rates ranging from 10% to 90%.
>
> Regarding refresh period, we wouldn't expect methods to be highly sensitive to this parameter as it only determines how many recently seen samples are considered for modeling the score distribution for percentile selection. *We also conducted an additional ablation study* varying $\tau$ from 50 to 300 (default was 100) on two datasets following the setting in Section 5.3. As the table below shows, the results remain consistent across different $\tau$ values.
>
> |Method|F101 (τ=50)|F101 (τ=200)|F101 (τ=300)|F101-N (τ=50)|F101-N (τ=200)|F01-N (τ=300)|
> |---|---|---|---|---|---|---|
> |PEAKS|**73.0 (±1.4)**|**72.8 (±1.3)**|**72.3 (±1.1)**|**62.0 (±2.6)**|**62.4 (±2.5)**|**61.9 (±2.4)**|
> |Moderate|70.4 (±1.3)|70.6 (±1.2)|71.1 (±1.2)|61.8 (±2.7)|62.2 (±3.4)|61.8 (±2.9)|
> |EL2N |66.3 (±1.4)|66.8 (±1.5)|67.5 (±1.8)|44.4 (±2.3)| 45.4 (±2.5)|45.4 (±2.7)|
> |Uncertainty|71.3 (±1.4)|71.8 (±1.9)|72.1 (±1.8)|54.6 (±2.7)|55.3 (±2.2)|55.7 (±3.0)|
>
> **Weakness-6**
>
> In Section 3, we assume that after the initialization phase (training on a few random samples), our pre-trained model acquires "decent" performance. We will clarify in the revised paper that this is far from satisfactory performance. To illustrate this clearly, we provide *additional results* below, comparing the accuracy of the pre-trained model after the initialization phase, and after IDS using PEAKS (budget x4). As shown, the model's performance after initialization does not approach the performance we report after IDS
>
> |Dataset|After Init|After IDS|
> |---|---|---|
> |C100|41.7 ±4.0|82.7 ±1.1|
> |F101|37.8 ±1.1|72.9 ±1.4|
> |F101-N|24.9 ±2.0|62.2 ±3.3|
> |WebVision|26.9 ±3.9|59.0 ±0.4|

---

### Official Review · Reviewer_YBNe · 2025-03-14

**Overall Recommendation:** 4

**Summary:**

This work introduces Incremental Data Selection (IDS) and proposes PEAKS, a method that selects training samples based on prediction error and kernel similarity. PEAKS efficiently builds training datasets while improving model performance. Experiments show it outperforms existing methods, significantly reducing data needs while maintaining accuracy, especially on large-scale datasets like WebVision.

**Claims And Evidence:**

This paper supports its claims clearly with a combination of mathematical derivations and detailed experimental setups.

**Essential References Not Discussed:**

There are no essential references missing in the paper.

**Experimental Designs Or Analyses:**

The experimental design appears detailed and supports the reliability of the conclusions.

**Methods And Evaluation Criteria:**

The proposed methods and evaluation criteria are meaningful for incremental data selection. PEAKS is well-suited for streaming data, and benchmark datasets ensure comprehensive evaluation.

**Other Comments Or Suggestions:**

None.

**Other Strengths And Weaknesses:**

Strengths:
1: The idea is explained clearly and is also innovative.
2: The experiments are thorough, making the conclusions highly convincing.

**Questions For Authors:**

The Neural Tangent Kernel seems to be applicable only to infinitely wide networks. Are there alternative methods for other networks?

**Relation To Broader Scientific Literature:**

PEAK combines similarity and uncertainty, offering an innovative approach not widely explored in existing methods, advancing related research.

**Theoretical Claims:**

The paper does not involve highly complex mathematical derivations. The presented formulas are correctly applied and support the method’s claims. No issues were found with their usage.

---

> ### Author Rebuttal · Authors · 2025-03-31
>
> We sincerely appreciate the reviewer's supportive and positive evaluation of our work.
>
> **The Neural Tangent Kernel seems to be applicable only to infinitely wide networks. Are there alternative methods for other networks?**
>
> The reviewer raises an important clarification point regarding the Neural Tangent Kernel (NTK) being theoretically guaranteed only for infinitely wide networks. We acknowledge that our networks do not operate in the strict NTK regime. However, based on intuition that in the model fine-tuning phase, weight changes happen less dramatically, our approach uses a first-order Taylor approximation as a practical tool to model network behavior between consecutive updates, and is not claiming the network exists in the kernel regime. [1] also used first-order approximations to derive methods and shed insight for practical networks like ours. Our empirical results also support this position.
>
> [1] Sketchy Moment Matching: Toward Fast and Provable Data Selection for Finetuning, NeurIPS 2024

---

### Official Review · Reviewer_Cx9C · 2025-03-14

**Overall Recommendation:** 3

**Summary:**

This paper poses Incremental Data Selection (IDS) problem where examples arrive continuously during training. Then it proposes a prediction error-based method PEAKS to address the problem, showing that a sample’s impact is influenced by both its position in feature space and its prediction error. Experimental results demonstrate the effectiveness of proposed method compared to naive baselines (such as random selection and Moderate).

**Claims And Evidence:**

Yes

**Essential References Not Discussed:**

No

**Experimental Designs Or Analyses:**

Yes

**Methods And Evaluation Criteria:**

Yes

**Other Comments Or Suggestions:**

The reviewer suggests that the authors add a summary of the paper's contributions at the end of the introduction. This summary would help emphasize the key points and guide readers through the rest of the manuscript.

**Other Strengths And Weaknesses:**

Strengths
1. Incremental Data Selection problem is interesting and seems to be important in real world applications.
2. The paper is well organized and written.

Weakness
1. It is unclear why and how the proposed method (PEAKS) is better than other baselines in IDS setting. The experimental results have verified the effectiveness of PEAKS, while further explanation and demonstration is needed.
2. How the proposed method performs on standard offline data selection setting, can it still outperforms other baselines?
3. The experimental setup is not aligned with existing works. For classification like CIFAR-100, a standard setup is SGD
optimizer with a momentum of 0.9, weight decay of 5e-4, and an initial learning rate of 0.1. While this work adopts adamw optimizer, which achieves worse performance than SGD on classification datasets. It remains unclear if the proposed method performs other baselines when using SGD optimizer.
4. Lack of experiments on large-scale real-world datasets like ImageNet-1K.
5. This paper focuses on online setting and the authors argue that the scenario naturally extends to continual learning
when the input distribution evolves over time. Therefore, it is helpful to conduct experiments on online continual learning benchmarks to demonstrate the performance, e.g., can PEAKS enhances the ER [1] and DER++ [2] by replacing random selection with PEAKS.
6. The reviewer suggests a simple baseline: selecting training samples which are wrongly predicted but with lower confidences (maximum posterior probability).

[1] Learning to learn without forgetting by maximizing transfer and minimizing interference.

[2] Dark Experience for General Continual Learning: a Strong, Simple Baseline.

**Questions For Authors:**

No

**Relation To Broader Scientific Literature:**

Fair

**Theoretical Claims:**

No theoretical claims

---

> ### Author Rebuttal · Authors · 2025-03-31
>
> We thank the reviewer for their thorough feedback.
>
> **Weakness-1**
>
> We thank the reviewer for pointing out that the draft lacks discussion of why PEAKS is effective. In the revised manuscript, we will explicitly clarify this. We believe PEAKS' main advantage is its ability to discriminate hard examples from outliers and noise. By combining two scores, it effectively picks typical but hard examples. In contrast, error-based methods like EL2N and GraNd purely focus on high-error samples that might be mislabeled. Moderate selection is conservative, filtering out noise but also ignoring valuable examples. Uncertainty is an unsupervised metric that does not exploit label information, making it immune to label noise but also limiting the information it can leverage.
>
> **Weakness-2**
>
> *We ran some additional experiments in the offline setting.* We first trained a ResNet-18 on the full Food101-N dataset (our largest dataset for training from scratch experiments, focusing on one dataset due to time limitations). Second, we used PEAKS and other baselines to rank the full dataset based on scores. Finally, we trained a model from scratch using 50%, 60%, 70%, and 80% of the top samples.  As seen from the results below (across 3 seeds), PEAKS performs best across most of the data budgets.
> ||50%|60%|70%|80%|
> |---|---|---|---|---|
> |Random|70.1 (±0.1)|71.1 (±0.1)|72.4 (±0.1)|73.6 (±0.1)|
> |Moderate|**71.0** (±2.0)|72.2 (±1.6)|73.6 (±1.1)|74.9 (±0.9)|
> |EL2N|64.9 (±4.9)|69.2 (±2.3)|71.9 (±0.9)|73.8 (±0.2)|
> |Uncertainty|66.5 (±3.5)|69.7 (±1.1)|71.9 (±0.2)|73.5 (±0.3)|
> |PEAKS|70.5 (±1.5)|**72.3** (±1.1)|**74.1** (±1.3)|**76.0** (±0.1)|
>
> While these preliminary results are promising, we want to emphasize that PEAKS was designed for the incremental setting, where an example's value is tied to the current model state. Offline data selection has different constraints and opportunities, such as the ability to globally optimize selection across the entire dataset.
>
> **Weakness-3**
>
> We ran a small grid search to select the optimizer, lr, and weight decay that optimized performance during the initial training phase (Appendix B). This approach was motivated by two key considerations:
>
> 1. In realistic streaming scenarios, hyperparameters must be selected based on early performance signals rather than retrospectively after seeing final results.
> 2. The initial phase is method-independent. This ensures a fair comparison where no method is advantaged by the optimizer choice.
>
> We found AdamW to perform better in our ResNet-18 and WebVision experiments, likely due to differences from the typical epoch-based full-dataset training. For ViT experiments, we used SGD with momentum (lr 0.001), which aligns with lr options explored by the original ViT authors. Notably, they do not consider a high lr such as 0.1 for fine-tuning (see Appendix Table 4 of [1]). Thus, PEAKS demonstrates strong performance with both AdamW and SGD.
>
> [1] An Image is Worth 16x16 Words: Transformers for Image Recognition at Scale, ICLR 2021
>
> **Weakness-4**
>
> We would like to highlight that the WebVision dataset contains 2.4 million images across 1000 classes, making it larger than ImageNet-1K. Furthermore, PEAKS shows the most promising results on this large real-world dataset. As explained in Appendix B.5, we deliberately avoided datasets closely related to ImageNet to prevent a potential bias, since ImageNet serves as our pretraining source.
>
> **Weakness-5**
>
> We thank the reviewer for this insightful suggestion. Studying IDS under distribution shift would indeed be interesting. However, we would like to clarify that we propose IDS as a problem setting, where data arrives incrementally but from a relatively stable distribution. While we noted that the scenario extends to continual learning (CL), this was to acknowledge the relationship rather than claim PEAKS would directly transfer to that setting without modification. PEAKS was designed to select samples that lead to maximum change in logits. However, such samples may not necessarily be good for avoiding catastrophic forgetting or providing knowledge transfer, which are crucial for CL. CL would likely need different considerations. Extending our work to CL would be a significant research direction requiring substantial modifications. We've noted this as important future work.
>
>
> **Weakness-6**
>
> Thank you for suggesting this baseline. *We implemented it exactly as described*: by assigning a score of 0 to correctly predicted examples and a score of 1 - max(softmax) to incorrect ones. Results below replicate our setting from Section 5.2.
>
> ||C100|F101|F101-N|WebVision|
> |---|---|---|---|---|
> |Data x1|62.7 (±7.0)|48.7 (±3.1)|35.0 (±3.7)|35.2 (±0.4)|
> |Data x2|77.0 (±2.8)|60.3 (±2.9)|44.8 (±3.7)|40.8 (±0.1)|
> |Data x4|83.9 (±1.0)|70.7 (±2.3)|53.5 (±3.8)|45.3 (±0.3)|
>
> Compared with Table-2 in our paper, this baseline performs strongly on CIFAR100, on par with GraNd. However, on the three larger datasets, it significantly lags behind PEAKS.

---

### Official Review · Reviewer_WR8r · 2025-03-15

**Overall Recommendation:** 3

**Summary:**

This paper focuses on data selection of DNNs that data arrives as a continuous stream and must be selected without access to the full data source. Based on this, the incremental data selection (IDS) problem is formulated as a three-stage process which including initialization with random samples, (streaming) data selection with model update and final training. To resolve IDS problem, this paper analyzes the impact of new data on model and proposes a score function based on the prediction error and penultimate representation similarity for measuring data importance. As a result, a new algorithm PEAKS is proposed. Experiments on CIFAR-100, FOOD101, Webvision demonstrate the effectiveness of PEAKS compared to random selection, embedding similarity based method, EL2N and GraNd.

**Claims And Evidence:**

The claims about IDS and PEAKS are clear and supported by convincing evidence.

**Essential References Not Discussed:**

To the best of my knowledge, essential references are included

**Experimental Designs Or Analyses:**

Lack of analysis of time complexity

**Methods And Evaluation Criteria:**

The proposed method (i.e., PEAKS) makes sense but the evaluation can be further improved to compare their empirical time complexity.

**Other Comments Or Suggestions:**

None

**Other Strengths And Weaknesses:**

Strength:
1. Incremental data selection is a novel setting for data efficient learning
2. The paper is well structured and easy to follow

Weakness:
1. IDS still needs to finetune the model on the selected data subset, which does not follow a purely streaming setting.
2. Lack of experimental evidence for the rationality of approximation in deriving scoring function
3. Lack of analysis about the empirical time complexity w.r.t. IDS and PEAKS

**Questions For Authors:**

My concerns are listed in the weakness section

**Relation To Broader Scientific Literature:**

The idea of incremental data selection is also related to online/continual learning

**Theoretical Claims:**

I have checked the theoretical analysis in section 3. The derivation is correct but the claim is not well supported. Approximations are introduced, including replacing the Jacobian w.r.t. the network parameters with parameters only in the last layer and replacing the mean embeddings of validation data with the weight vector. However, there lacks analysis of how close the approximated scoring function (i.e., Eq. 7 or Eq. 11) compares to the original one (Eq. 3).

---

> ### Author Rebuttal · Authors · 2025-03-31
>
> We are grateful to the reviewer for their time and constructive comments.
>
> **Lack of analysis about the empirical time complexity**
>
> We thank the reviewer for highlighting this concern. We will clarify in the main text that all baselines are subject to similar time complexity constraints. All methods require only a forward pass to make selection decisions—with GraNd being the only exception, as it requires an additional backward pass. Furthermore, we fixed the selection rate of IDS constant for all baselines (e.g., 20\% in main experiments). Thus, the sample acquisition speed is consistent across all baselines.
>
> **There lacks analysis of how close the approximated scoring function (i.e., Eq.7 or Eq.11) compares to the original one (Eq.3).**
>
> We agree with the reviewer's concern and have conducted an additional analysis to address this point.
>
> Our intuition for using last layer weight updates as an indicator for the whole network is supported by several studies in feature learning/transfer learning: early layers are more transferable [1, 2]; with pre-training, shallower layers approach optimality faster [3]; and early layers undergo less structural change than later ones during training [4]. Therefore, late layers in comparison are subjected to a lot of changes, thus can be representative of the whole network changes.
>
> To quantify the amount of change analytically is out of the scope of this paper. Additionally, note that explicitly computing Eq.3 requires calculating the Jacobian for every validation sample, which is of size (number of parameters × number of classes). Due to computational constraints, *we performed the following analysis on a toy MNIST scenario* rather than on large datasets and the ViT architecture.
>
> We first trained a two-hidden-layer MLP with 400 neurons each on 5k MNIST examples for 10 epochs. This step replicates a pretrained model that was trained on a few examples at the initialization phase to achieve some performance on the underlying task. Next, we held out 500 examples for validation and considered another 500 samples for selection using 3 methods:
> 1. Computing Eq.3 exactly.
> 2. Eq.9 (last layer training assumption).
> 3. Eq.11 (last layer training assumption and avoiding validation set).
>
> We used a 20% selection following our setting in the paper.
>
> Across 10 runs with different seeds, we found high overlap between selected subsets: $\mathbf{98 (\pm 1.4)}$\% between Eq.3 and Eq.9, and $\mathbf{92.9 (\pm 0.9)}$\% between Eq.3 and Eq.11. This demonstrates that although we make some simplifying assumptions, our approximate selection mechanisms closely match decisions with exact computation. We are committed to expanding this analysis and scaling it to more realistic networks and datasets such as CIFAR10 and ResNet-18 in the revised paper as computational resources permit.
>
> [1] How transferable are features in deep neural networks?, NeurIPS 2014.
>
> [2] What is being transferred in transfer learning?, NeurIPS 2020.
>
> [3] Which Layer is Learning Faster? A Systematic Exploration of Layer-wise Convergence Rate for Deep Neural Networks, ICLR 2023.
>
> [4] Unveiling the Dynamics of Transfer Learning Representations, ICLR 2024 Workshop Re-Align.
>
> **IDS still needs to finetune the model on the selected data subset, which does not follow a purely streaming setting.**
>
> We would like to clarify the goal of this final training phase and believe that in our writing, calling it "fine-tuning" might have led to misrepresentation. This final training phase is optional and we added it to ensure that the model also has a chance to iterate a few times over samples selected toward the end of data selection so that final accuracy is representative of the corresponding dataset size. However, this only provides an incremental improvement in performance. Below we report the performance of PEAKS before this final training and after. As seen from these results, PEAKS can also work on purely streaming performance with a minimal accuracy drop due to recently selected examples not yet being learned.
>
> | **Dataset** | **Before Final Training** | **After Final Training** |
> |----------------------|:---------------------------:|:--------------------------:|
> | CIFAR100 (×1)        | 57.56  | 58.99 |
> | CIFAR100 (×2)        | 70.03| 72.73|
> | CIFAR100 (×4)        | 80.71| 82.74|
> | Food101 (×1)         | 49.88| 50.85 |
> | Food101 (×2)         | 60.55| 62.64 |
> | Food101 (×4)         | 69.39| 72.88|
> | Food101N (×1)        | 40.71| 41.42|
> | Food101N (×2)        | 50.92 | 52.63|
> | Food101N (×4)        | 59.16| 62.16|
> | WebVision (×1)       | 37.75| 43.92|
> | WebVision (×2)       | 47.33 | 53.11|
> | WebVision (×4)       | 54.53| 59.02|
>
> *Average over three seeds.*

---

### Decision · Program_Chairs · 2025-05-01

**Decision:**

Accept (poster)

**Comment:**

The paper introduces a novel IDS problem and proposes an effective method, PEAKS. It shows strong performance in experiments but lacks generalizability and detailed analysis on computational overhead and hyperparameter sensitivity. Overall, it’s a promising contribution with room for improvement.